# Context-Dependent Upper-Confidence Bounds for Directed Exploration

**Raksha Kumaraswamy**[1], **Matthew Schlegel**[1], **Adam White**[1,2], **Martha White**[1]

[1]Department of Computing Science, University of Alberta; [2]DeepMind

{kumarasw, mkschleg}@ualberta.ca, adamwhite@google.com, whitem@ualberta.ca

## Abstract

Directed exploration strategies for reinforcement learning are critical for learning an optimal policy in a minimal number of interactions with the environment. Many algorithms use optimism to direct exploration, either through visitation estimates or upper-confidence bounds, as opposed to data-inefficient strategies like $\epsilon$-greedy that use random, undirected exploration. Most data-efficient exploration methods require significant computation, typically relying on a learned model to guide exploration. Least-squares methods have the potential to provide some of the data-efficiency benefits of model-based approaches—because they summarize past interactions—with the computation closer to that of model-free approaches. In this work, we provide a novel, computationally efficient, incremental exploration strategy, leveraging this property of least-squares temporal difference learning (LSTD). We derive upper-confidence bounds on the action-values learned by LSTD, with context-dependent (or state-dependent) noise variance. Such context-dependent noise focuses exploration on a subset of variable states, and allows for reduced exploration in other states. We empirically demonstrate that our algorithm can converge more quickly than other incremental exploration strategies using confidence estimates on action-values.

## 1  Introduction

Exploration is crucial in reinforcement learning, as the data gathering process significantly impacts the optimality of the learned policies and values. The agent needs to balance the amount of time taking exploratory actions to learn about the world, versus taking actions to maximize cumulative rewards. If the agent explores insufficiently, it could converge to a suboptimal policy; exploring too conservatively, however, results in many suboptimal decisions. The goal of the agent is *data-efficient exploration*: to minimize how many samples are wasted in exploration, particularly exploring parts of the world that are known, while still ensuring convergence to the optimal policy.

To achieve such a goal, directed exploration strategies are key. Undirected strategies, where random actions are taken such as in $\epsilon$-greedy, are a common default. In small domains these methods are guaranteed to find an optimal policy [35], because the agent is guaranteed to visit the entire space—but may take many many steps to do so, as undirected exploration can interfere with improving policies in incremental control. In this paper we explore the idea of constructing confidence intervals around the agent's value estimates. The agent can use these learned confidence intervals to select actions with the highest upper-confidence bound ensuring actions selected are of high value or whose values are highly uncertain. This optimistic approach is promising for directed exploration, but as yet there are few such methods that are model-free, incremental and computationally efficient.

Directed exploration strategies have largely been explored under the framework of "optimism in the face of uncertainty" [13]. These can generally be categorized into count-based approaches and confidence-based approaches. Count-based approaches estimate the "known-ness" of a state,

typically by maintaining counts for finite state-spaces [16, 6, 36, 37, 43] and extensions on counting for continuous states [14, 10, 26, 19, 33, 15, 32, 21]. Confidence interval estimates, on the other hand, depend on variance of the target, not just on visitation frequency for states. Confidence-based approaches can be more data-efficient for exploration, because the agent can better direct exploration where the estimates are less accurate. The majority of confidence-based approaches compute confidence intervals on model parameters, both for finite state-spaces [12, 47, 16, 6, 2, 3, 9, 43, 29] and continuous state-spaces [11, 27, 8, 1, 28]. There is early work quantifying uncertainty for value estimates directly for finite state-spaces [22], describing the difficulties with extending the local measures of uncertainty from the bandit literature to RL, since there are long-term dependencies.

These difficulties suggest why using confidence intervals directly on value estimates for exploration in RL has been less explored, until recently. More approaches are now being developed that maintain confidence intervals on the value function for continuous state-spaces, by maintaining a distribution over value functions [8, 31], or by maintaining a randomized set of value functions from which to sample [46, 31, 30, 34, 25]. Though significant steps forward, these approaches have limitations particularly in terms of computational efficiency. Delayed Gaussian Process Q-learning (DGPQ) [8] requires updating two Gaussian processes, which is cubic in the number of basis vectors for the Gaussian process. RLSVI [31] is relatively efficient, maintaining a Gaussian distribution over parameters with Thompson sampling to get randomized values. Their staged approach for finite-horizon problems, however, does not allow for value estimates to be updated online, as the value function is fixed per episode to gather an entire trajectory of data. Moerland et al. [25], on the other hand, sample a new parameter vector from the posterior distribution each time an action is considered, which is expensive. The bootstrapping approaches can be efficient, as they simply have to store several value functions, either for training on a bootstrapped subset of samples—such as in Bootstrapped DQN [30]—or for maintaining a moving bootstrap around the changing parameters themselves, for UCBootstrap [46]. For both of these approaches, however, it is unclear how many value functions would be required, which could be large depending on the problem.

In this paper, we provide an incremental, model-free exploration algorithm with fast converging upper-confidence bounds, called UCLS: Upper-Confidence Least-Squares. We derive the upper-confidence bounds for Least-Squares Temporal Difference learning (LSTD), taking advantage of the fact that LSTD has an efficient summary of past interaction to facilitate computation of confidence intervals. Importantly, these upper-confidence bounds have context-dependent variance, where variance is dependent on state rather than a global estimate, focusing exploration on states with higher-variance. Computing confidence intervals for action-values in RL has remained an open problem, and we provide the first theoretically sound result for obtaining upper-confidence bounds for policy evaluation under function approximation, without making strong assumptions on the noise. We demonstrate in several simulated domains that UCLS outperforms DGPQ, UCBootstrap, and RLSVI. We also empirically show the benefit of using UCLS to a simplified version that uses a global variance estimate, rather than context-dependent variance.

## 2  Background

We focus on the problem of learning an optimal policy for a Markov decision process, from on-policy interaction. A Markov decision process consists of $(\mathcal{S}, \mathcal{A}, \Pr, r, \gamma)$ where $\mathcal{S}$ is the set of states; $\mathcal{A}$ is the set of actions; $\Pr : \mathcal{S} \times \mathcal{A} \times \mathcal{S} \to [0, \infty)$ provides the transition probabilities; $r : \mathcal{S} \times \mathcal{A} \times \mathcal{S} \to \mathbb{R}$ is the reward function; and $\gamma : \mathcal{S} \times \mathcal{A} \times \mathcal{S} \to [0, 1]$ is the transition-based discount function which enables either continuing or episodic problems to be specified [45]. On each step, the agent selects action $A_t$ in state $S_t$, and transitions to $S_{t+1}$, according to $\Pr$, receiving reward $R_{t+1} \stackrel{\text{def}}{=} r(S_t, A_t, S_{t+1})$ and discount $\gamma_{t+1} \stackrel{\text{def}}{=} \gamma(S_t, A_t, S_{t+1})$. For a policy $\pi : \mathcal{S} \times \mathcal{A} \to [0, 1]$, where $\sum_{a \in \mathcal{A}} \pi(s, a) = 1 \ \forall s \in \mathcal{S}$, the value at a given state $s$, taking action $a$, is the expected discounted sum of future rewards, with actions selected according to $\pi$ into the future,

$$Q^\pi(s, a) = \mathbb{E}\Big[R_{t+1} + \gamma_{t+1} \sum_{a \in \mathcal{A}} \pi(S_{t+1}, a) Q^\pi(S_{t+1}, a) \Big| S_t = s, A_t = a\Big]$$

For problems in which $Q^\pi$ can be stored in a table, a fixed point for the action-values $Q^\pi$ exists for a given $\pi$. In most domains, $Q^\pi$ must be approximated by $Q^\pi_{\mathbf{w}}$, parametrized by $\mathbf{w} \in \mathcal{W} \subset \mathbb{R}^d$.

In the case of linear function approximation, state-action features $\mathbf{x}(s_t, a_t)$ are used to approximate action-values $Q^\pi_{\mathbf{w}}(s_t, a_t) = \mathbf{x}(s_t, a_t)^\top \mathbf{w}$. The weights $\mathbf{w}$ can be learned with a stochastic approximation algorithm, called temporal difference (TD) learning [39]. The TD update [39] processes

samples one at a time, $\mathbf{w} = \mathbf{w} + \alpha\delta_t\mathbf{z}_t$, with $\delta_t \stackrel{\text{def}}{=} R_{t+1} + \gamma_{t+1}\mathbf{x}_{t+1}^\top\mathbf{w} - \mathbf{x}_t^\top\mathbf{w}$ for $\mathbf{x}_t \stackrel{\text{def}}{=} \mathbf{x}(S_t, A_t)$. The eligibility trace $\mathbf{z}_t = \mathbf{x}_t + \gamma_{t+1}\lambda\mathbf{z}_{t-1}$ facilitates multi-step updates via an exponentially weighted memory of previous feature activations decayed by $\lambda \in [0, 1]$ and $\mathbf{z}_0 = \mathbf{0}$. Alternatively, we can directly compute the weight vector found by TD using least-squares temporal difference learning (LSTD) [5]. The LSTD solution is more data-efficient, and can avoid the need to tune TD's stepsize parameter $\alpha > 0$. The LSTD update can be efficiently computed incrementally without approximation or storing the data [5, 4], by maintaining a matrix $\mathbf{A}_T$ and vector $\mathbf{b}_T$,

$$\mathbf{A}_T \stackrel{\text{def}}{=} \frac{1}{T}\sum_{t=0}^{T-1}\mathbf{z}_t(\mathbf{x}_t - \gamma_{t+1}\mathbf{x}_{t+1})^\top \qquad \mathbf{b}_T \stackrel{\text{def}}{=} \frac{1}{T}\sum_{t=0}^{T-1}\mathbf{z}_t R_{t+1} \qquad (1)$$

The value function approximation at time step $T$ is the weights that satisfy the linear system $\mathbf{A}_T\mathbf{w} = \mathbf{b}_T$. In practice, the inverse of the matrix $\mathbf{A}^{-1}$ is maintained using a Sherman-Morrison update, with a small regularizer $\eta$ added to the matrix $\mathbf{A}$ to guarantee invertibility [41].

One approach to ensure systematic exploration is to initialize the agent's value estimates optimistically. The action-value function must be initialized to predict the maximum possible return (or greater) from each state and action. For example, for cost-to-goal problems, with -1 per step, the values can be initialized to zero. For continuing problems, with constant discount $\gamma_c < 1$, the values can be initialized to $G_{max} = R_{max}/(1 - \gamma_c)$, if the maximum reward $R_{max}$ is known. For fixed features that are non-negative and encode locality—such as tile coding or radial basis functions—the weights $\mathbf{w}$ can be simply set to $G_{max}$, to make $Q_{\mathbf{w}}$ optimistic.

More generally, however, it can be problematic to use optimistic initialization. Optimistic initialization assumes the beginning of time is special—a period when systematic exploration should be performed after which the agent should more or less exploit its current knowledge. Many problems are non-stationary—or at least benefit from a tracking approach due to aliasing caused by function approximation—and benefit from continual exploration. Further, unlike for fixed features, it is unclear how to set and maintain initial values at $G_{max}$ for learned features, such as with neural networks. Optimistic initialization is also not straightforward for algorithms like LSTD, which completely overwrite the estimate $\mathbf{w}$ on each step with a closed-form solution. In fact, we have found that this issue with LSTD has been obfuscated, because the regularizer $\eta$ has inadvertently played a role in providing optimism (see Appendix A). Rather, to use optimism in LSTD for control, we need to explicitly compute upper-confidence bounds.

Confidence intervals around action-values, then, provide another mechanism for exploration in reinforcement learning. Consider action selection with explicit confidence intervals around mean estimates $\hat{Q}_{\mathbf{w}}(S_t, A_t)$, with estimated radius $\hat{U}(S_t, A_t)$. The action selection is greedy w.r.t. to these optimistic values, $\text{argmax}_a \hat{Q}_{\mathbf{w}}(S_t, a) + \hat{U}(S_t, a)$, which provides a high-confidence upper bound on the best possible value for that action. The use of upper-confidence bounds on value estimates for exploration has been well-studied and motivated theoretically in online learning [7]. In reinforcement learning, there have only been a few specialized proofs for particular algorithms using optimistic estimates [8, 31], but the result can be expressed more generally by using the idea of stochastic optimism. We extract the central argument by Osband et al. [31] to provide a general Optimistic Values Theorem in Appendix B. In particular, similar to online learning, we can guarantee that greedy-action selection according to upper-confidence values will converge to the optimal policy, if the confidence interval radius shrinks to zero, if the algorithm to estimate action-values for a policy converges to the corresponding actions and if upper-confidence estimates are stochastically optimal—remain above the optimal action-values in expectation.

Motivated by this result, we pursue principled ways to compute upper-confidence bounds for the general, online reinforcement learning setting. We make a step towards computing such values incrementally, under function approximation, by providing upper-confidence bounds for value estimates made by LSTD, for a fixed policy. We approximate these bounds to create a new algorithm for control—called Upper-Confidence-Least-Squares (UCLS).

## 3  Estimating Upper-Confidence Bounds for Policy Evaluation using LSTD

Consider the goal of obtaining a confidence interval around value estimates learned incrementally by LSTD for a fixed policy $\pi$. The value estimate is $\mathbf{x}^\top\mathbf{w}$ for state-action features $\mathbf{x}$ for the current state and action. We would like to guarantee, with probability $1 - p$ for a small $p > 0$, that the confidence

interval around this estimate contains the value $\mathbf{x}^\top \mathbf{w}^*$ given by the optimal $\mathbf{w}^* \in \mathcal{W}$. To estimate such an interval without parametric assumptions, we use Chebyshev's inequality which—unlike other concentration inequalities like Hoeffding or Bernstein—does not require independent samples.

To use this inequality, we need to determine the variance of the estimate $\mathbf{x}^\top \mathbf{w}$; the variance of the estimate, given $\mathbf{x}$, is due to the variance of the weights. Let $\mathbf{w}^*$ be fixed point solution for the projected Bellman operator for the $\lambda$-return—the TD fixed point, for a fixed policy $\pi$. To characterize the noise for this optimal estimator, let $\nu_t$ be the TD-error for the optimal weights $\mathbf{w}^*$, where

$$r_{t+1} = (\mathbf{x}_t - \gamma \mathbf{x}_{t+1})^\top \mathbf{w}^* + \nu_t \qquad \text{with } \mathbb{E}[\nu_t \mathbf{z}_t] = 0. \tag{2}$$

The expectation is taken across all states weighted by the sampling distribution, typically the stationary distribution $\mathbf{d}_\pi : \mathcal{S} \to [0, \infty)$ or in the off-policy case the stationary distribution of the behaviour policy. We know that $\mathbb{E}[\nu_t \mathbf{z}_t] = 0$, by the definition of the Projected Bellman Error fixed point.

This noise $\nu_t$ is incurred from the variability in the reward, the variability in the transition dynamics and potentially the capabilities of the function approximator. A common assumption—when using linear regression for contextual bandits [20] and for reinforcement learning [31]—is that the variance of the target is a constant value $\sigma^2$ for all contexts $\mathbf{x}$. Such an assumption, however, is likely to produce larger confidence intervals than necessary. For example, consider a one-state world with two actions, where one action has a high variance reward and the other has a lower variance reward (see Appendix A, Figure 4). A global sample variance will encourage both actions to be taken many times. For data-efficient exploration, however, the agent should take the high-variance action more, and only needs a few samples from the low-variance action.

We derive a confidence interval for LSTD, in Theorem 1. We also derive the confidence interval assuming a global variance in Corollary 1, to provide a comparison. We compare to using this global-variance upper-confidence bound in our experiments, and show that it results in significantly worse performance than using a context-dependent variance. Note that we do not assume $\mathbf{A}_T$ is invertible; if we did, the big-O term in (3) below would disappear. We include this term for preciseness of the result—even though we will not estimate it—because for smaller $T$, $\mathbf{A}_T$ is unlikely to be invertible. However, we expect this big-O term to get small quickly, and be dominated by the other terms. In our algorithm, therefore, we ignore the big-O term.

**Theorem 1.** *Let $\bar{\nu}_T \overset{\text{def}}{=} \frac{1}{T} \sum_{t=0}^{T-1} \mathbf{z}_t \nu_t$ and $\mathbf{w}_T = \mathbf{A}_T^+ \mathbf{b}_T$ where $\mathbf{A}_T^+$ is the pseudoinverse of $\mathbf{A}_T$. Let $\boldsymbol{\epsilon}_T^* \overset{\text{def}}{=} (\mathbf{A}_T^+ \mathbf{A}_T - \mathbf{I})\mathbf{w}^*$ reflect the degree to which $\mathbf{A}_T$ is not invertible; it is zero when $\mathbf{A}_T$ is invertible. Assume that the following are all finite: $\mathbb{E}[\mathbf{A}_T^+ \bar{\nu}_T + \boldsymbol{\epsilon}_T^*]$, $\mathbb{V}[\mathbf{A}_T^+ \bar{\nu}_T + \boldsymbol{\epsilon}_T^*]$ and all state-action features $\mathbf{x}$. With probability at least $1 - p$, given state-action features $\mathbf{x}$,*

$$\mathbf{x}^\top \mathbf{w}^* \leq \mathbf{x}^\top \mathbf{w}_T + \sqrt{\tfrac{p+1}{p}} \sqrt{\mathbf{x}^\top \mathbb{E}[\mathbf{A}_T^+ \bar{\nu}_T \bar{\nu}_T^\top \mathbf{A}_T^{+\top}]\mathbf{x}} + O\left(\mathbb{E}[(\mathbf{x}^\top \boldsymbol{\epsilon}_T^*)^2]\right) \tag{3}$$

**Proof:** First we compute the mean and variance for our learned parameters. Because $r_{t+1} = (\mathbf{x}_t - \gamma \mathbf{x}_{t+1})^\top \mathbf{w}^* + \nu_t$,

$$
\begin{aligned}
\mathbf{w}_T &= \left( \frac{1}{T} \sum_{t=0}^{T-1} \mathbf{z}_t (\mathbf{x}_t - \gamma \mathbf{x}_{t+1})^\top \right)^{-1} \left( \frac{1}{T} \sum_{t=0}^{T-1} \mathbf{z}_t r_{t+1} \right) \\
&= \mathbf{A}_T^+ \left( \frac{1}{T} \sum_{t=0}^{T-1} \mathbf{z}_t ((\mathbf{x}_t - \gamma \mathbf{x}_{t+1})^\top \mathbf{w}^* + \nu_t) \right) \\
&= \mathbf{A}_T^+ \mathbf{A}_T \mathbf{w}^* + \mathbf{A}_T^+ \left( \frac{1}{T} \sum_{t=0}^{T-1} \mathbf{z}_t \nu_t \right) \\
&= \mathbf{w}^* + \mathbf{A}_T^+ \bar{\nu}_T + \boldsymbol{\epsilon}_T^*
\end{aligned}
$$

This estimate has a small amount of bias, that vanishes asymptotically. But, for a finite sample,

$$\mathbb{E}\left[ \mathbf{A}_T^+ \left( \frac{1}{T} \sum_{t=0}^{T-1} \mathbf{z}_t \nu_t \right) \right] \neq \mathbb{E}[\mathbf{A}_T^+] \mathbb{E}\left[ \frac{1}{T} \sum_{t=0}^{T-1} \mathbf{z}_t \nu_t \right] = \mathbf{0}.$$

Further, because $\mathbf{A}_T$ may not be invertible, there is an additional error $\boldsymbol{\epsilon}_T^*$ term which will vanish with enough samples, i.e., once $\mathbf{A}_T$ can be guaranteed to be invertible.

For covariance, because

$$\mathbf{w}_T - \mathbb{E}[\mathbf{w}_T] = \left(\mathbf{w}^* + \mathbf{A}_T^+ \bar{\boldsymbol{\nu}}_T + \boldsymbol{\epsilon}_T^*\right) - \mathbb{E}\left[\mathbf{w}^* + \mathbf{A}_T^+ \bar{\boldsymbol{\nu}}_T + \boldsymbol{\epsilon}_T^*\right)]$$
$$= \mathbf{A}_T^+ \bar{\boldsymbol{\nu}}_T + \boldsymbol{\epsilon}_T^* - \mathbb{E}\left[\mathbf{A}_T^+ \bar{\boldsymbol{\nu}}_T + \boldsymbol{\epsilon}_T^*\right]$$

the covariance of the weights is

$$\mathbb{V}[\mathbf{w}_T] = \mathbb{V}\left[\mathbf{A}_T^+ \bar{\boldsymbol{\nu}}_T + \boldsymbol{\epsilon}_T^*\right]$$

The goal for computing variances is to use a concentration inequality. Chebyshev's inequality[1] states that for a random variable $X$, if the $\mathbb{E}[X]$ and $\mathbb{V}[X]$ are bounded, then for any $\epsilon \geq 0$:

$$\mathrm{Pr}\left(|X - \mathbb{E}[X]| < \epsilon \sqrt{\mathbb{V}[X]}\right) \geq 1 - \frac{1}{\epsilon^2}$$

If we set $\epsilon = \sqrt{1/p}$, then this gives

$$\mathrm{Pr}\left(|X - \mathbb{E}[X]| < \sqrt{\tfrac{1}{p}} \sqrt{\mathbb{V}[X]}\right) \geq 1 - p$$

Now we have characterized the variance of the weights, but what we really want is to characterize the variance of the value estimates. Notice that the variance of the value-estimate, for state-action $\mathbf{x}$ is

$$\mathbb{V}[\mathbf{x}^\top \mathbf{w}_T | \mathbf{x}] = \mathbb{E}[\mathbf{x}^\top \mathbf{w}_T \mathbf{w}_T^\top \mathbf{x} | \mathbf{x}] - \mathbb{E}[\mathbf{x}^\top \mathbf{w}_t | \mathbf{x}]^2$$
$$= \mathbf{x}^\top \left(\mathbb{E}[\mathbf{w}_T \mathbf{w}_T^\top] - \mathbb{E}[\mathbf{w}_T]\mathbb{E}[\mathbf{w}_T]^\top\right) \mathbf{x}$$
$$= \mathbf{x}^\top \mathbb{V}[\mathbf{w}_T] \mathbf{x}$$

Therefore, the variance of the estimate is characterized by the variance of the weights. With high probability,

$$\left|\mathbf{x}^\top \mathbf{w}_T - \mathbf{x}^\top \mathbf{w}^*\right| = \left|\mathbf{x}^\top (\mathbf{w}_T - \mathbb{E}[\mathbf{w}_T]) + \mathbf{x}^\top (\mathbb{E}[\mathbf{w}_T] - \mathbf{w}^*)\right|$$
$$\leq \left|\mathbf{x}^\top (\mathbf{w}_T - \mathbb{E}[\mathbf{w}_T])\right| + \left|\mathbf{x}^\top (\mathbb{E}[\mathbf{w}_T] - \mathbf{w}^*)\right|$$
$$\leq \frac{1}{\sqrt{p}} \sqrt{\mathbf{x}^\top \mathbb{V}\left[\mathbf{A}_T^+ \bar{\boldsymbol{\nu}}_T + \boldsymbol{\epsilon}_T^*\right] \mathbf{x}} + \left|\mathbf{x}^\top \mathbb{E}[\mathbf{A}_T^+ \bar{\boldsymbol{\nu}}_T + \boldsymbol{\epsilon}_T^*]\right| \quad (4)$$
$$= \frac{1}{\sqrt{p}} \sqrt{\mathbf{x}^\top \left(\mathbb{E}\left[\mathbf{A}_T^+ \bar{\boldsymbol{\nu}}_T \bar{\boldsymbol{\nu}}_T^\top \mathbf{A}_T^{+\top} + \boldsymbol{\Sigma}_T^*\right] - \boldsymbol{\mu}_T^* \boldsymbol{\mu}_T^{*\top}\right) \mathbf{x}} + \sqrt{\mathbf{x}^\top \boldsymbol{\mu}_T^* \boldsymbol{\mu}_T^{*\top} \mathbf{x}} \quad (5)$$

where Equation 4 uses Chebyshev's inequality, and the last step is a rewriting of Equation 4 using the definitions $\boldsymbol{\mu}_T^* \overset{\text{def}}{=} \mathbb{E}[\mathbf{A}_T^+ \bar{\boldsymbol{\nu}}_T + \boldsymbol{\epsilon}_T^*]$ and $\boldsymbol{\Sigma}_T^* \overset{\text{def}}{=} \mathbf{A}_T^+ \bar{\boldsymbol{\nu}}_T \boldsymbol{\epsilon}_T^{*\top} + \boldsymbol{\epsilon}_T^* (\mathbf{A}_T^+ \bar{\boldsymbol{\nu}}_T)^\top + \boldsymbol{\epsilon}_T^* \boldsymbol{\epsilon}_T^{*\top}$.

To simplify (5), we need to determine an upper bound for the general formula $c\sqrt{a^2 - b^2} + b$ where $a \geq b \geq 0$. Because $p < 1$, we know that $c = \sqrt{1/p} \geq 1$. Therefore, the extremal points for $b$, $b = a$ and $b = 0$, both result in an upper bound of $ca$. Taking the derivative of the objective, gives a single stationary point in-between $[0, a]$, with $b = \frac{a}{\sqrt{c^2+1}}$. The value at this point evaluates to be $a\sqrt{c^2 + 1}$. Therefore, this objective is upper-bounded by $a\sqrt{c^2 + 1}$.

Now for $a^2 = \mathbf{x}^\top \mathbb{E}\left[\mathbf{A}_T^+ \bar{\boldsymbol{\nu}}_T \bar{\boldsymbol{\nu}}_T^\top \mathbf{A}_T^{+\top} + \boldsymbol{\Sigma}_T^*\right] \mathbf{x}$, the term involving $\mathbf{x}^\top \mathbb{E}[\boldsymbol{\Sigma}_T^*] \mathbf{x}$ should quickly disappear, since it is only due to the potential lack of invertibility of $\mathbf{A}_T$. This term is equal to $\mathbb{E}\left[2(\mathbf{x}^\top \mathbf{A}_T^+ \bar{\boldsymbol{\nu}}_T)(\mathbf{x}^\top \boldsymbol{\epsilon}_T^*) + (\mathbf{x}^\top \boldsymbol{\epsilon}_T^*)^2\right]$, which results in the additional $O(\mathbb{E}[(\mathbf{x}^\top \boldsymbol{\epsilon}_T^*)^2])$ in the bound. ∎

**Corollary 1.** *Assume that $\nu_t$ are i.i.d., with mean zero and bounded variance $\sigma^2$. Let $\bar{\mathbf{z}}_T = \frac{1}{T}\sum_{t=0}^{T-1} \mathbf{z}_t$ and assume that the following are finite: $\mathbb{E}[\boldsymbol{\epsilon}_T^*]$, $\mathbb{V}[\boldsymbol{\epsilon}_T^*]$, $\mathbb{E}[\mathbf{A}_T^+ \bar{\mathbf{z}}_T \bar{\mathbf{z}}_T^\top \mathbf{A}_T^{+\top}]$ and all state-action features $\mathbf{x}$. With probability at least $1 - p$, given state-action features $\mathbf{x}$,*

$$\mathbf{x}^\top \mathbf{w}^* \leq \mathbf{x}^\top \mathbf{w}_T + \sigma \sqrt{\tfrac{p+1}{p}} \sqrt{\mathbf{x}^\top \mathbb{E}[\mathbf{A}_T^+ \bar{\mathbf{z}}_T \bar{\mathbf{z}}_T^\top \mathbf{A}_T^{+\top}]\mathbf{x}} + O\left(\mathbb{E}[(\mathbf{x}^\top \boldsymbol{\epsilon}_T^*)^2]\right) \quad (6)$$

**Proof:** The result follows similarly to above, with some simplifications due to global-variance:

$$\mathbb{E}\left[\mathbf{A}_T^+\bar{\boldsymbol{\nu}}_T\right] = \mathbb{E}\left[\mathbb{E}\left[\mathbf{A}_T^+\bar{\boldsymbol{\nu}}_T\Big|S_0,....,S_T\right]\right] = \mathbb{E}\left[\mathbf{A}_T^+\frac{1}{T}\sum_{t=0}^{T-1}\mathbf{z}_t\mathbb{E}\left[\nu_t\Big|S_0,....,S_T\right]\right] = \mathbf{0}$$

$$\mathbb{E}[\mathbf{A}_T^+\bar{\boldsymbol{\nu}}_T\bar{\boldsymbol{\nu}}_T^\top\mathbf{A}_T^{+\top}] = \sigma^2\mathbb{E}[\mathbf{A}_T^+\bar{\mathbf{z}}_T\bar{\mathbf{z}}_T^\top\mathbf{A}_T^{+\top}]$$

∎

## 4   UCLS: Estimating upper-confidence bounds for LSTD in control

In this section, we present Upper-Confidence-Least-Squares (UCLS)[2], a control algorithm, which incrementally estimates the upper-confidence bounds provided in Theorem 1, for guiding on-policy exploration. The upper-confidence bounds are sound without requiring i.i.d. assumptions; however, they are derived for a fixed policy. In control, the policy is slowly changing, and so instead we will be slowly tracking this upper bound. The general strategy, like policy iteration, is to slowly estimate both the value estimates and the upper-confidence bounds, under a changing policy that acts greedily with respect to the upper-confidence bounds. Tracking these upper bounds incurs some approximations; we identify and address potential issues here. The complete psuedocode for UCLS is given in the Appendix (Algorithm 2).

First, we are not evaluating one fixed policy; rather, the policy is changing. The estimates $\mathbf{A}_T$ and $\mathbf{b}_T$ will therefore be out-of-date. As is common for LSTD with control, we use an exponential moving average, rather than a sample average, to estimate $\mathbf{A}_T$, $\mathbf{b}_T$ and the upper-confidence bound. The exponential moving average uses $\mathbf{A}_T = (1-\beta)\mathbf{A}_{T-1} + \beta\mathbf{z}_T(\mathbf{x}_t - \gamma\mathbf{x}_{t+1})^\top$, for some $\beta \in [0,1]$. If $\beta = 1/T$, then this reduces to the standard sample average; otherwise, for a fixed $\beta$, such as $\beta = 0.01$, more recent samples have a higher weight in the average. Because an exponential average is unbiased, the result in Theorem 1 would still hold, and in practice the update will be more effective for the control setting.

Second, we cannot obtain samples of the noise $\nu_t = r_{t+1} + \gamma_{t+1}\mathbf{x}_{t+1}^\top\mathbf{w}^* - \mathbf{x}_t^\top\mathbf{w}^*$, which is the TD-error for the optimal value function parameters $\mathbf{w}^*$ (see Equation (2)). Instead, we use $\delta_t$ as a proxy. This proxy results in an upper bound that is too conservative—too loose—because $\delta_t$ is likely to be larger than $\nu_t$. This is likely to ensure sufficient exploration, but may cause more exploration than is needed. The moving average update

$$\bar{\boldsymbol{\nu}}_t = \bar{\boldsymbol{\nu}}_{t-1} + \beta_t(\delta_t\mathbf{z}_t - \bar{\boldsymbol{\nu}}_{t-1}) \tag{7}$$

should also help mitigate this issue, as older $\delta_t$ are likely larger than more recent ones.

Third, the covariance matrix $\mathbf{C}$ estimating $\mathbb{E}[\mathbf{A}_T^{-1}\bar{\boldsymbol{\nu}}_T\bar{\boldsymbol{\nu}}_T^\top\mathbf{A}_T^{-1}]$ could underestimate covariances, depending on a skewed distribution over states and depending on the initialization. This is particularly true in early learning, where the distribution over states is skewed to be higher near the start state; a sample average can result in underestimates in as yet unvisited parts of the space. To see why, let $\mathbf{a} = \mathbf{A}_T^{-1}\bar{\boldsymbol{\nu}}_T$. The covariance estimate $\mathbf{C}_{ij} = \mathbb{E}[\mathbf{a}_i\mathbf{a}_j]$ corresponds to feature $i$ and $j$. The agent begins in a certain region of the space, and so features that only become active outside of this region will be zero, providing samples $\mathbf{a}_i\mathbf{a}_j = 0$. As a result, the covariance is artificially driven down in unvisited regions of the space, because the covariance accumulates updates of 0. Further, if the initialization to the covariance $\mathbf{C}_{ii}$ is an underestimate, a visited state with high variance will artificially look more optimistic than an unvisited state.

We propose two simple approaches to this issue: updating $\mathbf{C}$ based on locality and adaptively adjusting the initialization to $\mathbf{C}_{ii}$. Each covariance estimate $\mathbf{C}_{ij}$ for features $i$ and $j$ should only be updated if the sampled outer-product is relevant, with the agent in the region where $i$ and $j$ are active. To reflect this locality, each $\mathbf{C}_{ij}$ is updated with the $\mathbf{a}_i\mathbf{a}_j$ only if the eligibility traces is non-zero for $i$ and $j$. To adaptively update the initialization, the maximum observed $\mathbf{a}_i^2$ is stored, as $c_{\max}$, and the initialization $c_0$ to each $\mathbf{C}_{ii}$ is retroactively updated using

$$\mathbf{C}_{ii} = \mathbf{C}_{ii} - (1-\beta)^{c_i}c_0 + (1-\beta)^{c_i}c_{\max}$$

where $c_i$ is the number of times $\mathbf{C}_{ii}$ has been updated. This update is equivalent to having initialized $\mathbf{C}_{ii} = c_{\max}$. We provide a more stable retroactive update to $\mathbf{C}_{ii}$, in the pseudocode in Algorithm 2, that is equivalent to this update.

Fourth, to improve the computational complexity of the algorithm, we propose an alternative, incremental strategy for estimating $\mathbf{w}$, that takes advantage of the fact that we already need to estimate the inverse of $\mathbf{A}$ for the upper bound. In order to do so, we make use of the summarized information in $\mathbf{A}$ to improve the update, but avoid directly computing $\mathbf{A}^{-1}$ as it may be poorly conditioned. Instead, we maintain an approximation $\mathbf{B} \approx \mathbf{A}^{-\top}$ that uses a simple gradient descent update, to minimize $\|\mathbf{A}^\top \mathbf{B} \mathbf{x}_t - \mathbf{x}_t\|_2^2$. If $\mathbf{B}$ is the inverse of $\mathbf{A}^\top$, then this loss is zero; otherwise, minimizing it provides an approximate inverse. This estimate $\mathbf{B}$ is useful for two purposes in the algorithm. First, it is clearly needed to estimate the upper-confidence bound. Second, it also provides a pre-conditioner for the iterative update $\mathbf{w} = \mathbf{w} + \mathbf{G}(\mathbf{b} - \mathbf{A}\mathbf{w})$, for preconditioner $\mathbf{G}$. The optimal preconditioner is in fact the inverse of $\mathbf{A}$, if it exists. We use $\mathbf{G} = \mathbf{B}^\top + \eta \mathbf{I}$ for a small $\eta > 0$ to ensure that the preconditioner is full rank. Developing this stable update for LSTD required significant empirical investigation into alternatives; in addition to providing a more practical UCLS algorithm, we hope it can improve the use of LSTD in other applications.

# 5   Experiments

We conducted several experiments to investigate the benefits of UCLS' directed exploration against other methods that use confidence intervals for action selection, to evaluate sensitivity of UCLS's performance with respect to its key parameter $p$, and to contrast the advantage contextual variance estimates offer over global variance estimates in control. Our experiments were intentionally conducted in small—though carefully selected—simulation domains so that we could conduct extensive parameter sweeps, hundreds of runs for averaging, and compare numerous state-of-the-art exploration algorithms (many of which are computationally expensive on larger domains). We believe that such experiments constitute a significant contribution, because effectively using confidence bounds for model free-exploration in RL is still in its infancy—not yet at the large-scale demonstration state–with much work to be done. This point is highlighted nicely below as we demonstrate that several recently proposed exploration methods fail on these simple domains.

## 5.1   Algorithms

We compare UCLS to DGPQ [8], UCBootstrap [46], our extension of LSPI-Rmax to an incremental setting [19] and RLSVI [31]. In-depth descriptions of each algorithm and implementation details can be found in the Appendix. These algorithms are chosen because they either keep confidence intervals explicitly, as in UCBootstrap, or implicitly as in DGPQ and RLSVI. In addition, we included LSPI-Rmax as a natural alternative approach to using LSTD to maintain optimistic value estimates.

We also include Sarsa with $\epsilon$-greedy, with $\epsilon$ optimized over an extensive parameter sweep. Though $\epsilon$-greedy is not a generally practical algorithm, particularly in larger worlds, we include it as a baseline. We do not include Sarsa with optimistic initialization, because even though it has been a common heuristic, it is not a general strategy for exploration. Optimistic initialization can converge to suboptimal solutions if initial optimism fades too quickly [46]. Further, initialization only happens once, at the beginning of learning. If the world changes, then an agent relying on systematic exploration due to its initialization may not react, because it no longer explores. For completeness comparing to previous work using optimistic initialization, we include such results in Appendix G.

## 5.2   Environments

**Sparse Mountain Car** is a version of classic mountain car problem Sutton and Barto [40], only differing in the reward structure. The agent only receives a reward of $+1$ at the goal and 0 otherwise, and a discounted, episodic $\gamma$ of 0.998. The start state is sampled from the range $[-0.6, -0.4]$ with velocity zero. This domain is used to highlight how exploration techniques perform when the reward signal is sparse, and thus initializing the value function to zero is not optimistic.

**Puddle World** is a continuous state 2-dimensional world with $(x, y) \in [0, 1]^2$ with 2 puddles: (1) $[0.45, 0.4]$ to $[0.45, 0.8]$, and (2) $[0.1, 0.75]$ to $[0.45, 0.75]$ - with radius 0.1 and the goal is the region $(x, y) \in ([0.95, 1.0], [0.95, 1.0])$. The agent receives a reward of $-1 - 400 * d$ on each time step, where $d$ denotes the distance between the agent's position and the center of the puddle, and an undiscounted, episodic $\gamma$ of 1.0. The agent can select an action to move $0.05 + \zeta$, $\zeta \sim N(\mu = 0, \sigma^2 = 0.01)$.

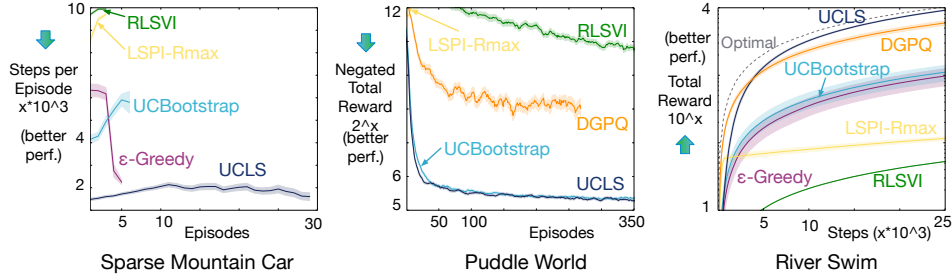

Figure 1: A comparison of speed of learning in Sparse Mountain Car, Puddle World and River Swim. In plots (a) and (b) lower on y-axis are better, whereas in (c) curves higher along y-axis are better. Sparse Mountain Car and Puddle World are episodic problems with a fixed experience budget. Thus the length of the lines in plots (a) and (b) indicate how many episodes each algorithm completed over 50,000 steps, and the height on the y-axis indicates the quality of the learned policy—lower indicates better performance. Note RLSVI did not show significant learning after 50,000 steps. The RLSVI result in Puddle World uses a budget of 1 million.

The agent's initial state is uniformly sampled from $(x, y) \in ([0.1, 0.3], [0.45, 0.65])$. This domain highlights a common difficulty for traditional exploration methods: high magnitude negative rewards, which often cause the agent to erroneously decrease its value estimates too quickly.

**River Swim** is a standard continuing exploration benchmark [42] inspired by a fish trying to swim upriver, with high reward (+1) upstream which is difficult to reach and, a lower but still positive reward (+0.005), which is easily reachable downstream. We extended this domain to continuous states in $[0, 1]$, with a stochastic displacement of $0.1$ when taking an action up or down, with low-probability of success for up. The starting position is sampled uniformly in $[0, 0.1]$, and $\gamma = 0.99$.

## 5.3 Experimental Setup

We investigate a learning regime where the agents are allowed a fixed budget of interaction steps with the environment, rather than allowing a finite number of episodes of unlimited length. Our primary concern is early learning performance, thus each experiment is restricted to 50,000 steps, with an episode cutoff (in Sparse Mountain Car and Puddle World) at 10,000 steps. In this regime, an agent that spends a significant time exploring the world during the first episode may not be able to complete many episodes, the cutoff makes exploration easier given the strict budget on experience. Whereas, in the more common framework of allowing a fixed number of episodes, an agent can consume many steps during the first few episodes exploring, which is difficult to detect in the final performance results. We average over 100 runs in River Swim and 200 runs for the other domains . For all the algorithms that utilize eligibility traces we set $\lambda$ to be 0.9. For algorithms which use exponential averaging, $\beta$ is set to 0.001, and the regularizer $\eta$ is set to be 0.0001. The parameters for UCLS are fixed. RLSVI's weights are recalculated using all experienced transitions at the beginning of an episode in Puddle World and Sparse Mountain Car, and every 5,000 steps in River Swim. The parameters of competitors, where necessary, are selected as the best from a large parameter sweep.

All the algorithms except DGPQ use the same representation: (1) Sparse Mountain Car - 8 tilings of 8x8, hashed to a memory space of 512, (2) River Swim - 4 tilings of granularity 32, hashed to a memory space of 128, and (3) Puddle World - 5 tilings of granularity 5x5, hashed to a memory space of 128. DGPQ uses its own kernel-based representation with normalized state information.

## 5.4 Results & Analysis

Our first experiment simply compares UCLS against other control algorithms in all the domains. Figure 1 shows the early learning results across all three domains. In all three domains UCLS achieves the best final performance. In Sparse Mountain Car, UCLS learns faster than the other methods, while in River Swim DGPQ learns faster initially. UCBootstrap and UCLS learn at a similar rate in Puddle World, which is a cost-to-goal domain. UCBootstrap, and bootstrapping approaches generally, can suffer from insufficient optimism, as they rely on sufficiently optimistic or diverse initialization strategies [46, 30]. LSPI-Rmax and RLSVI do not perform well in any of the domains. DGPQ does not perform as well as UCLS in Puddle World, and exhibits high variance compared with the other methods. In Puddle World, UCLS goes on to finish 1200 episodes in the alloted budget of steps,

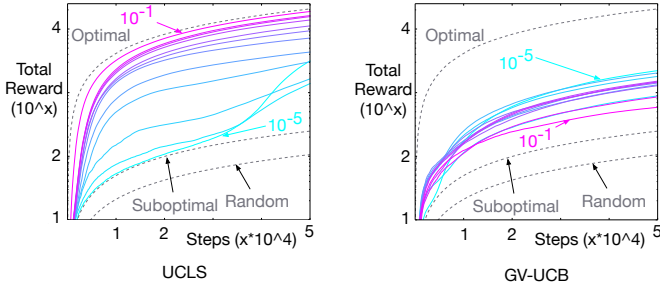

Figure 2: The effect of the confidence parameter $p$ on the policy, in River Swim, using context-dependent variance (UCLS) and global variance (GV-UCB). The values for $p$ are $\{10^{-5}, [1, 2, \ldots, 9] \times 10^{-3}, 10^{-2}, 10^{-1}\}$.

whereas in River Swim both UCLS and DGPQ get close to the optimal policy by the end of the experiment.

The DGPQ algorithm uses the maximum reward (Rmax) to initialize the Gaussian processes. In Sparse Mountain Car this effectively converts the problem back into the traditional -1 per-step formulation. In this traditional variant of Mountain Car UCLS significantly outperforms DGPQ (Appendix G). Sarsa with $\epsilon$-greedy learns well in Puddle world as it is a cost-to-goal problem in which by default Sarsa uses optimistic initialization, and therefore is reported in the Appendix. .

Next we investigated the impact of the confidence level $1 - p$, on the performance of UCLS in River Swim. The confidence interval radius is proportional to $\sqrt{1 + 1/p}$; smaller $p$ should correspond to a higher rate of exploration. In Figure 2, smaller $p$ resulted in a slower convergence rate, but all values eventually reach the optimal policy.

Finally, we investigate the benefit using contextual variance estimates over global variance estimates within UCLS. In Figure 2, we also show the effect of various $p$ values on the performance of the algorithm resulting from Corollary 1, which we call Global Variance-UCB (GV-UCB) (see Appendix E.1 for more details about this algorithm). For this range of $p$, UCLS still converges to the optimal policy, albeit at different rates. Using a global variance estimates (GV-UCB), on the other hand, results in significant over-estimates of variance, resulting in poor performance.

# 6 Conclusion and Discussion

This paper develops a sound upper-confidence bound on the value estimates for least-squares temporal difference learning (LSTD), without making i.i.d. assumptions about noise distributions. In particular, we allow for context-dependent noise, where variability could be due to noise in rewards, transition dynamics or even limitations of the function approximator. We then introduce an algorithm, called UCLS, that estimates these upper-confidence bounds incrementally, for policy iteration. We demonstrate empirically that UCLS requires far fewer exploration steps to find high-quality policies compared to several baselines, across domains chosen to highlight different exploration difficulties.

The goal of this paper is to provide an incremental, model-free, data-efficient, directed exploration strategy. The upper-confidence bounds for action-values for fixed policies are one of the few available under function approximation, and so a step towards exploration with optimistic values in the general case. A next step is to theoretically show that using these upper bounds for exploration ensures stochastic optimism, and so converges to optimal policies.

One promising aspect of UCLS is that it uses least-squares to efficiently summarize past experience, but is not tied to a specific state representation. Though we considered a fixed representation for UCLS, it is feasible that an analysis for the non-stationary case could be used as well for the setting where the representation is being adapted over time. If the representation drifts slowly, then UCLS may be able to similarly track the upper-confidence bounds. Recent work has shown that combining deep Q-learning with Least-squares can result in significant performance gains over vanilla DQN[18]. We expect that combining deep networks and UCLS could result in even larger gains, and is a natural direction for future work.

# 7 Acknowledgements

We would like to thank Bernardo Ávila Pires and Jian Qian for their helpful comments, alongwith Calcul Québec (`www.calculquebec.ca`) and Compute Canada (`www.computecanada.ca`) for the computing resources used in this work.

## Footnotes

[1]Bernstein's inequality cannot be used here because we do not have independent samples. Rather, we characterize behaviour of the random variable $\mathbf{w}$, using variance of $\mathbf{w}$, but cannot use bounds that assume $\mathbf{w}$ is the sum of independent random variables. The bound with Chebyshev will be loose, but we can better control the looseness of the bound with the selection of $p$ and the constant in front of the square root.

[2]We do not characterize the regret of UCLS, and instead similarly to policy iteration, rely on a sound update under a fixed policy to motivate incrementally estimating these values as if the policy is fixed and then acting according to them. The only model-free algorithm that achieves a regret bound is RLSVI, but that bound is restricted to the finite horizon, batch, tabular setting. It would be a substantial breakthrough to provide such a regret bound, and is beyond the scope of this work.

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
