[Supplementary Material · full.pdf]

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

Figure 3: Learning performance in Mountain Car for LSTD-in and LSTD-out with $\eta$ kept constant through learning (-C) and $\eta$ fading with time (-F). (a) Early learning curves for LSTD-in. This plot does not include LSTD-out as it performed too poorly to be visible. (b) Learning curves for LSTD-in with best and worst runs. LSTD-in-C's worst run performed too poorly to be visible. (c) Parameter sensitivity for both variants LSTD-in and LSTD-out to $\eta/\eta_r$.

# A  Issues with LSTD for control

LSTD is a more data-efficient algorithm than its incremental counterpart TD, and typically performs quite well in policy evaluation. This is primarily due to TD only using each sample once for a stochastic update with a tuned stepsize parameter. In the case of control, LSTD performs surprisingly well without $\epsilon$-greedy exploration and lack of an optimism strategy. We highlight here the inadvertent use of the regularization parameter as a form of optimism for LSTD in control, and empirically show when this strategy fails leading us to UCLS as a sound approach in using LSTD in control.

In practice, the inverted matrix $\mathbf{A}^{-1}$ is often directly maintained using a Sherman-Morrison update, with a small regularizer $\eta$ added to the matrix $\mathbf{A}$ to guarantee invertibility [41].

There are two objectives that can be solved when dealing with an ill-conditioned system $\mathbf{A}\mathbf{w} = \mathbf{b}$. The most common is to use Tikohonov regularization solving, referred to here as *LSTD-out*.

$$\min_{\mathbf{w}} \|\mathbf{A}\mathbf{w} - \mathbf{b}\|_2^2 + \eta_r \|\mathbf{w}\|_2^2$$

Another approach is to solve the system

$$\min_{\mathbf{w}} \|(\mathbf{A} + \eta\mathbf{I})\mathbf{w} - \mathbf{b}\|_2^2$$

The second approach is implicitly what is solved when a Sherman-Morrison update is used for $\mathbf{A}^{-1}$, with a small regularizer $\eta$ added to the matrix $\mathbf{A}$ to guarantee invertibility. This approach is referred to here as *LSTD-in*. When $\eta = 0$, both approaches are solving $\|\mathbf{A}\mathbf{w} - \mathbf{b}\|_2^2$, which may have infinitely many solutions if $\mathbf{A}$ is not full rank. While the Tikohonov regularization strategy is more common, the second approach is useful for enabling use of the incremental Sherman-Morrison update to facilitate maintaining $\mathbf{A}^{-1}$ directly.

Another choice in regularizing the ill-conditioned system is in how $\eta$ decays over time. A small fixed $\eta$ can be used as a constant regularizer, even as the number of samples increases, because the true $\mathbf{A}$ may be ill-conditioned. However, more regularization could also be used at the beginning and then decayed over time. The incremental Sherman-Morrison update implicitly decays $\eta$ proportionally to $\frac{1}{t}$.

We conducted an empirical study using LSTD without an $\epsilon$-greedy exploration strategy in two domains: Mountain Car and a new One-State world. One-State world—depicted in Figure 4— simulates a typical setting where sufficient exploration is needed: one outcome with low variance and lower expected value and one outcome with high variance and higher expected value. For an algorithm that does not explore sufficiently, it is likely to settle on the suboptimal action, but more immediately rewarding low-variance outcome. This world simulates a larger continuous navigation task from 46. We include results for both systems described above and consider a fading version (shown by *-F*) or a constant regularization parameter (shown by *-C*).

Figure 3 shows results for the four different LSTD strategies in Mountain Car. The Tikohonov regularization, with $\eta_r$, is unable to learn an optimal policy in this domain, whereas with either constant or fading $\eta$, the agent can learn an optimal policy. This is surprising, considering we use

$\mathbb{E}[R] = 1, \mathbb{V}[R] = 0$ $s$ $\mathbb{E}[R] = 2, \mathbb{V}[R] \approx 28$

Figure 4: One-state world, where the optimal action (right) has high-variance; the reward here is uniformly sampled from within the set $\{-5, -2, 2, 5, 10\}$. LSTD, with $\epsilon = 0$ and $\eta$ large, fails in this world, unlike the cost-to-goal problems.

Figure 5: $\eta$-sensitivity in 1-State world with various LSTD updates. Sarsa with optimistic initialization $\alpha = 0.001$ is used as a baseline. The y-axis represents percentage optimal behaviour, where optimal behaviour is choosing to go right, in 20k steps (averaged over 30 runs). Sarsa with optimistic initialization is highly sensitive to the step-size chosen. With other stepsizes (not shown in figure), it reduces its values too quickly, and fails a significant percentage of the time. The best stepsize is chosen here to show near-optimal performance is possible in the domain.

neither randomized exploration nor optimistic initialization. The parameter sensitivity curve, shown in plot c, indicates $\eta$ and $\eta_r$ needs to be sufficiently large as time passes in order to find an optimal policy.

Next, we show that neither regularization strategy with fading $\eta$ is effective in the One-State world. The optimal strategy is to take the Right action, to get an expected reward of 2 under a higher variance for obtaining rewards. All of the LSTD variants fail for this domain, because $\eta$ no longer plays a role in encouraging exploration. To verify that a directed exploration strategy helps, we experiment with $\epsilon$-greedy exploration, with $\epsilon = 0.1$, decayed by a factor of $0.2$ every 100 steps (shown in Figure 5). With $\epsilon$-greedy, and small values of $\eta_r$ and $\eta$, the policy converges to the optimal action, whereas it fails to with higher values of $\eta_r$ and $\eta$.

These results suggest that $\eta$'s role in exploration has obscured our understanding of how to use LSTD for control. LSTD, with sufficient optimism does seem to reach optimal solutions, and unlike Sutton et al. [38], we did not find any issues with forgetting. This further explains why there have been previous results with small $\epsilon$ for LSTD in cost-to-goal problems, that nonetheless still obtained the optimal policy [44]. Therefore, in developing UCLS, we more explicitly add optimism to LSTD, and ensure $\eta$ is strictly used as a regularization parameter (to ensure well-conditioned updates).

## B  Optimistic Values Theorem

The use of upper confidence bounds on value estimates for exploration has been well-studied and motivated theoretically in online learning [7]. For reinforcement learning, though, there are only specialized proofs for particular algorithms using optimistic estimates [8, 31]. To better motivate and appreciate the use of upper confidence bounds for reinforcement learning, we extract the key argument from Osband et al. [31], which uses the idea of stochastic optimism.

Under function approximation, it may not be possible to obtain the optimal policy exactly. Instead, our criterion is to obtain the optimal policy according to the following formulation, assuming greedy-action selection from action-values. Let $Q^* : \mathcal{S} \times \mathcal{A} \to \mathbb{R}$ be the action-values for the optimal policy,

under the chosen density $d : \mathcal{S} \times \mathcal{A} \to [0, \infty)$ over states and actions

$$Q^* = \underset{Q \in \mathcal{Q}}{\mathrm{argmax}} \int_{\mathcal{S} \times \mathcal{A}} d(s,a) Q(s,a) ds da \tag{8}$$

This optimization does not preclude $d$ being related to the trajectory of optimal policy, but generically allows specification of any density, such as one putting all weight on a set of start states or such as one that is uniform across states and actions to ensure optimality from any point in the space. The optimal policy in this setting is the policy that corresponds to acting greedily w.r.t. $Q^*$; depending on the function space $\mathcal{Q}$, this may only be an approximately optimal policy. The design of the agent is directed towards this goal, though we do not explicitly optimize this objective.

Let $\tilde{Q}_t = \hat{Q}_t + \hat{U}_t$ be the estimated action-values plus the confidence interval radius $\hat{U}_t$ on time step $t$, to get the estimated upper confidence bound which the agent uses to select actions. Let $\pi_t$ be the policy induced by greedy action selection on $\tilde{Q}_t$.

**Assumption 1** (Stochastic Optimism). *At some point $T > 0$, the action-values at every step $t \geq T$ are* stochastically optimistic*: $\mathbb{E}[\tilde{Q}_t(S,A)] \geq \mathbb{E}[Q^*(S,A)]$, with expectation according to a specified density $d : \mathcal{S} \times \mathcal{A} \to [0, \infty)$.*

**Assumption 2** (Shrinking Confidence Interval Radius). *The confidence interval radius $\hat{U}_t$ goes to zero: $\mathbb{E}[\hat{U}_t(S,A)] \leq f(t)$ for some non-negative function $f$ with $f(t) \to 0$.*

**Assumption 3** (Convergent Action Values). *The estimated action-values $\hat{Q}_t$ approach the true action-values for policy $\pi_t$: $\left| \mathbb{E}[\hat{Q}_t(S,A) - Q^{\pi_t}(S,A)] \right| \leq g(t)$ for some non-negative function $g$ with $g(t) \to 0$.*

These assumptions are heavily dependent on the distribution utilized to evaluate the expectation. If the expectations are w.r.t. the stationary distribution induced by the optimal policy ($d^*$), it is easy to see that they could be satisfied - as the density is non-zero only for the optimal state-action pairs. In contrast, if the density is a uniform density over the space, then these assumptions may not be satisfied.

Given the three key assumptions, the theorem below is straightforward to prove. However, these three conditions are fundamental, and do not imply each other. Therefore, this result highlights what would need to be shown, to obtain the Optimistic Values Theorem. For example, Assumption 1 and 2 do not imply Assumption 3, because the confidence interval radius could decrease to zero, and $\hat{Q}_t$ still be stochastically optimistic and an over-estimate of values that correspond to a suboptimal policy. Assumption 1 and 3 do not imply Assumption 2, because $\hat{Q}_t$ could converge to the policy corresponding to acting greedily w.r.t. $\tilde{Q}_t$, but $\hat{U}_t$ may never fade away. Then, $\tilde{Q}_t$ could still be stochastically optimistic, but the policy $\pi_t$ could be suboptimal because it is acting greedily according to inaccurate, inflated estimates of value $\tilde{Q}_t$.

**Theorem 2** (Optimistic Values Theorem). *Under Assumptions 1, 2 and 3,*

$$\mathbb{E}[Q^*(S,A)] - \mathbb{E}[Q^{\pi_t}(S,A)] \leq f(t) + g(t)$$

$$Regret(T) \overset{\text{def}}{=} \sum_{t=1}^{T} \mathbb{E}[Q^*(S,A)] - \mathbb{E}[Q^{\pi_t}(S,A)]$$

$$\leq \sum_{t=1}^{T} f(t) + g(t)$$

**Proof:** Consider the regret across states and actions

$$\mathbb{E}[Q^*(S,A) - Q^{\pi_t}(S,A)] = \mathbb{E}[Q^*(S,A) - \tilde{Q}_t(S,A)] + \mathbb{E}[\tilde{Q}_t(S,A) - Q^{\pi_t}(S,A)]$$
$$\leq \mathbb{E}[\tilde{Q}_t(S,A) - Q^{\pi_t}(S,A)]$$

because $\mathbb{E}[Q^*(S,A) - \tilde{Q}_t(S,A)] \leq 0$ by Assumption 1. By Assumptions 2 and 3,

$$\mathbb{E}[\tilde{Q}_t(S,A) - Q^{\pi_t}(S,A)] = \mathbb{E}[\hat{Q}_t(S,A) - Q^{\pi_t}(S,A)] + \mathbb{E}[\hat{U}_t(S,A)]$$
$$\leq g(t) + f(t)$$

completing the proof. ∎

This result is intentionally abstract, where the three assumptions could be satisfied in a variety of ways. These assumptions have been verified for one algorithm, called RLSVI, under a tabular setting using a finite-horizon specification [31], which simplifies ensuring stochastic optimism (Assumption 1). We hypothesize that the last two assumptions could be addressed with a two-timescale analysis, with confidence interval radius $\hat{U}_t$ updating more slowly than $\hat{Q}_t$. This would reflect an iterative approach, where the optimistic values are essentially held fixed—such as is done in Delayed Q-learning [8]—and $Q^{\pi_t}$ estimated, before then adjusting the optimistic values. The updates to $\hat{Q}_t$, then, would be updated on a faster timescale, converging to $Q^{\pi_t}$, and the upper confidence radius $\hat{U}_t$ updating on a slower timescale.

---

**Algorithm 1** GetOptimisticAction($\mathbf{x}_{s,\cdot}$)

---

$u_a \leftarrow \sqrt{\left(1 + \frac{1}{p}\right) \mathbf{x}_{s,a}^\top \mathbf{C} \mathbf{x}_{s,a}} \quad \forall a \in \mathcal{A}$

$a = \operatorname{argmax}_{a \in \mathcal{A}} \mathbf{x}_{s,a}^\top \mathbf{w} + u_a$

**return** $a$

---

**Algorithm 2** UCLS($\lambda$)

---

$\mathbf{A} \leftarrow \mathbf{0}, \mathbf{b} \leftarrow \mathbf{0}, \mathbf{z} \leftarrow \mathbf{0}, \mathbf{w} \leftarrow \mathbf{0}$

$\mathbf{B} \leftarrow \mathbf{I}, \mathbf{C} \leftarrow \mathbf{I}, \bar{\boldsymbol{\nu}} \leftarrow \mathbf{0}, \mathbf{c} \leftarrow \mathbf{1}$

$p = 0.1, \eta = 10^{-4}, \beta = 0.001, c_{\max} = 1.0$

$\mathbf{x}_{s,\cdot} \leftarrow$ initial state-action features, for any action

$a \leftarrow$ GetOptimisticAction($\mathbf{x}_{s,\cdot}$)

**repeat**

    Take action $a$ and observe $\mathbf{x}_{s',\cdot}$ and $r$, and $\gamma$

    $a' \leftarrow$ GetOptimisticAction($\mathbf{x}_{s',\cdot}$)

    $\delta \leftarrow r + (\gamma \mathbf{x}_{s',a'} - \mathbf{x}_{s,a})^\top \mathbf{w}$

    $\mathbf{z} \leftarrow \gamma \lambda \mathbf{z} + \mathbf{x}_{s,a}$

    $\mathbf{b} \leftarrow (1 - \beta)\mathbf{b} + \beta r \mathbf{z}$

    $\mathbf{A} \leftarrow (1 - \beta)\mathbf{A} + \beta \mathbf{z}(\mathbf{x}_{s,a} - \gamma \mathbf{x}_{s',a'})^\top$

    ▷ Update $\mathbf{B} \approx \mathbf{A}^{-\top}$

    $\alpha = \min\left\{1.0, \frac{0.01}{||\mathbf{A}||_F^2 ||\mathbf{x}_{s,a}||_2^2 + 1.0}\right\}$

    $\mathbf{B} \leftarrow \mathbf{B} - \alpha \mathbf{A}(\mathbf{A}^\top \mathbf{B} \mathbf{x}_{s,a} - \mathbf{x}_{s,a})\mathbf{x}_{s,a}^\top$

    ▷ Update $\mathbf{C}$

    $\bar{\boldsymbol{\nu}} \leftarrow (1 - \beta)\bar{\boldsymbol{\nu}} + \beta \delta \mathbf{z}$

    $\mathbf{a} \leftarrow \mathbf{B}^\top \bar{\boldsymbol{\nu}}$

    temp $= c_{\max}$

    $c_{\max} = \max(c_{\max}, \mathbf{a}_1^2, \ldots, \mathbf{a}_d^2)$

    **if** temp $\neq c_{\max}$ **then**                ▷ Adjust initialization

        $\mathbf{C}_{ii} \leftarrow \mathbf{C}_{ii} + \mathbf{c}_i(c_{\max} - \text{temp}), \forall i$

    **for** $i$ such that $\mathbf{z}_i \neq 0$ **do**

        $\mathbf{c}_i = \mathbf{c}_i(1 - \beta)$

        **for** $j$ such that $\mathbf{z}_j \neq 0$ **do**

            $\mathbf{C}_{ij} \leftarrow (1 - \beta)\mathbf{C}_{ij} + \beta \mathbf{a}_i \mathbf{a}_j$

    ▷ Update $\mathbf{w}$

    $\mathbf{w} \leftarrow \mathbf{w} + (\mathbf{B}^\top + \eta \mathbf{I})(\mathbf{b} - \mathbf{A}\mathbf{w})$

    $\mathbf{x}_{s,a} \leftarrow \mathbf{x}_{s',a'} \quad \text{and} \quad a \leftarrow a'$

**until** agent done interaction with environment

---

## C Estimating Upper Confidence Bounds for Policy Evaluation using linear TD

Recall that the TD update [39] processes one sample at a time as $\mathbf{w}_{t+1} = \mathbf{w}_t + \alpha\delta_t\mathbf{z}_t$ to estimate the solution to the least-squares system $\mathbf{w}_T = \mathbf{A}_T^{-1}\mathbf{b}_T$ in an incremental manner. This is feasible as the following holds:

$$\mathbf{w}_T = \mathbf{A}_T^{-1}\mathbf{b}_T$$
$$\mathbf{A}_T\mathbf{w}_T = \mathbf{b}_T$$
$$\left[\frac{1}{T}\sum_{t=0}^{T-1}\mathbf{z}_t(\mathbf{x}_t - \gamma_{t+1}\mathbf{x}_{t+1})^\top\right]\mathbf{w}_T = \left[\frac{1}{T}\sum_{t=0}^{T-1}\mathbf{z}_t r_t\right]$$
$$\sum_{t=0}^{T-1}\mathbf{z}_t(r_t + \gamma_{t+1}\mathbf{x}_{t+1}^\top\mathbf{w}_T - \mathbf{x}_t^\top\mathbf{w}_T) = 0$$
$$\sum_{t=0}^{T-1}\mathbf{z}_t\delta_t = 0$$

Therefore, $\mathbf{w}_t$ is updated incrementally with a constant step-size towards minimizing this error stochastically.

Given this incremental method to estimate a least-squares solution, we can notice that the covariance matrix is the outer-product of the solution to a similar least-squares system, $\mathbf{A}_T^{-1}\bar{\boldsymbol{\nu}}_T$. The solution to this least-squares system is denoted by $\mathbf{w}_{\mathrm{var}}$, and can be estimated incrementally as:

$$\mathbf{w}_{\mathrm{var}t+1} = \mathbf{w}_{\mathrm{var}t} + \alpha\delta_{\mathrm{var}t}\mathbf{z}_t$$

where, $\delta_{\mathrm{var}t} = \delta_t + \gamma_{t+1}\mathbf{x}_{t+1}^\top\mathbf{w}_{\mathrm{var}t} - \mathbf{x}_t^\top\mathbf{w}_{\mathrm{var}t}$.

Therefore, for a given policy, the true action-values satisfy the following:

$$\mathbf{x}^\top\mathbf{w}^* \leq \mathbf{x}^\top\mathbf{w}_T + \sqrt{\frac{p+1}{p}}\sqrt{\mathbf{x}^\top\mathbf{w}_{\mathrm{var}T}\mathbf{w}_{\mathrm{var}T}^\top\mathbf{x}}$$

Similarly a linear variant of GV-UCB can be obtained as the upper bound again consists of an outer-product to a different least-squares system $\mathbf{A}_T^{-1}\bar{\mathbf{z}}_T$. But as shown in Figures 2 and 7, GV-UCB, the quadratic version, can be highly sample inefficient, which may worsen with the linear variant, GV-UCB-L. Therefore, we do not provide an algorithm, or any empirical results for GV-UCB-L here.

## D UCLS-L: Estimating upper confidence bounds for linear TD in control

In the same spirit as UCLS utilizes the policy evaluation upper-bound of LSTD for control, with a slowly changing control policy, UCLS-L utilizes the policy evaluation upper-bound of linear TD for control. At each step, UCLS-L, given in Algorithm 4, uses a stochastic update to estimate mean action-values, and their corresponding contextual-variance estimates. These stochastic updates use fixed, and if necessary are different, step-sizes ($\alpha$, and $\alpha_{\mathrm{var}}$ respectively), instead of a closed-form solution as done by UCLS. The rate of change of the policy in UCLS-L is controlled by the step-size, unlike in UCLS which utilizes weighted forms of experience samples in $\mathbf{A}$ and $\mathbf{b}$. Therefore, UCLS-L can be sensitive to the step-sizes, but adapt more quickly to a changing feature-space. Further, in order to account for underestimates of variances, UCLS-L uses another vector $\mathbf{w}_{\mathrm{varInit}}$, in a similar spirit as UCLS's retroactive initialization of covariance estimates. Additionally, as these upper-bounds are estimated incrementally, they can be quite loose, specifically so in the linear framework. Therefore, instead of choosing the best parameter $p$, we can choose a parameter $\bar{p} = \sqrt{1 + \frac{1}{p}}$: the loss of theoritical interpretation of the upper-bound is traded-off for better empirical performance.

With this, we investigate UCLS-L as a substitue to UCLS in the three benchmark domains. For UCLS-L, both $p$ and $\bar{p}$ is swept, from which the best parameter is selected scale the uncertainty unstemiate, along with the learning rates $\alpha$ and $\alpha_{\mathrm{var}}$. The experiment configuration and the domains are the same as used in UCLS. The results are presented in Figure 6. UCLS-L does reasonably

**Algorithm 3** GetOptimisticActionLinear($\mathbf{x}_{s,\cdot}$)

---

$u_a \leftarrow \sqrt{\left(1 + \frac{1}{p}\right)\left((\mathbf{x}_{s,a}^\top \mathbf{w}_{\mathrm{var}})^2 + ||\mathbf{x}_{s,a}||^2_{\mathbf{I}\mathbf{w}_{\mathrm{varInit}}}\right)} \quad \forall a \in \mathcal{A}$

$a = \operatorname{argmax}_{a \in \mathcal{A}} \mathbf{x}_{s,a}^\top \mathbf{w} + u_a$

**return** $a$

---

**Algorithm 4** UCLS-L($\lambda$)

---

$p = 0.1$, $\beta = 0.001$, $v_{\mathrm{init}} = 1.0$, $\alpha = 0.01$, $\alpha_{\mathrm{var}} = 0.1$
$\mathbf{w} \leftarrow \mathbf{0}$, $\mathbf{w}_{\mathrm{var}} \leftarrow \mathbf{0}$, $\mathbf{w}_{\mathrm{varInit}} \leftarrow \mathbf{1} * v_{\mathrm{init}}$, $\mathbf{c} \leftarrow \mathbf{1}$
$\mathbf{x}_{s,\cdot} \leftarrow$ initial state-action features, for any action
$a \leftarrow$ GetOptimisticActionLinear($\mathbf{x}_{s,\cdot}$)
**repeat**
    Take action $a$ and observe $\mathbf{x}_{s',\cdot}$ and $r$, and $\gamma$
    $a' \leftarrow$ GetOptimisticActionLinear($\mathbf{x}_{s',\cdot}$)
    $\delta \leftarrow r + (\gamma \mathbf{x}_{s',a'} - \mathbf{x}_{s,a})^\top \mathbf{w}$
    $\delta_{\mathrm{var}} \leftarrow \delta + (\gamma \mathbf{x}_{s',a'} - \mathbf{x}_{s,a})^\top \mathbf{w}_{\mathrm{var}}$
    $\mathbf{z} \leftarrow \gamma \lambda \mathbf{z} + \mathbf{x}_{s,a}$
    ▷ Update $\mathbf{w}_{\mathrm{var}}$ and $\mathbf{w}_{\mathrm{varInit}}$
    $\mathbf{w}_{\mathrm{var}} \leftarrow \mathbf{w}_{\mathrm{var}} + \alpha_{\mathrm{var}} \delta_{\mathrm{var}} \mathbf{z}$
    temp $= v_{\mathrm{init}}$
    $v_{\mathrm{init}} = \max(v_{\mathrm{init}}, \mathbf{w}_{\mathrm{var}1}^2, \ldots, \mathbf{w}_{\mathrm{var}d}^2)$
    **if** temp $\neq v_{\mathrm{init}}$ **then**                       ▷ Adjust initialization
        $\mathbf{w}_{\mathrm{varInit}i} \leftarrow \mathbf{w}_{\mathrm{varInit}i} + \mathbf{c}_i(v_{\mathrm{init}} - \text{temp}), \forall i$
    **for** $i$ such that $\mathbf{z}_i \neq 0$ **do**
        $\mathbf{c}_i = \mathbf{c}_i(1 - \beta)$
        $\mathbf{w}_{\mathrm{varInit}i} \leftarrow (1 - \beta) * \mathbf{w}_{\mathrm{varInit}i}, \forall i$
    ▷ Update $\mathbf{w}$
    $\mathbf{w} \leftarrow \mathbf{w} + \alpha \delta \mathbf{z}$
    $\mathbf{x}_{s,a} \leftarrow \mathbf{x}_{s',a'}$   and   $a \leftarrow a'$
**until** agent done interaction with environment

---

well in all the domains. While it experiences more regret in Puddle World, and River Swim during early learning, by the end of the steps budget, it learns the optimal policy. In Sparse Mountain Car, surprisingly, UCLS-L learns much faster and a better policy than UCLS. This can be attributed to the fact that the parameter $p$ in UCLS was not swept, whereas in UCLS-L we did sweep to find the best parameter to scale the variance estimate. As the domain is a sparse-reward domain, the variance estimates play a significant role in influencing exploratory behaviour, and therefore optimizing for $p$ would improve UCLS' performance. Nonetheless, these results show UCLS-L to be a promising algorithm for linear complexity based control, and warrant further evaluation of it.

**Algorithm 5** GetOptimisticActionGlobal($\mathbf{x}_{s,\cdot}$)

---

$u_a \leftarrow \sigma \sqrt{\left(1 + \frac{1}{p}\right) \mathbf{x}_{s,a}^\top \mathbf{C} \mathbf{x}_{s,a}} \quad \forall a \in \mathcal{A}$

$a = \operatorname{argmax}_{a \in \mathcal{A}} \mathbf{x}_{s,a}^\top \mathbf{w} + u_a$

**return** $a$

---

# E  Details about other algorithms

## E.1  Global variance UCB

Based on Corollary 1 to estimate a global variance $\sigma^2$, it is possible that the noise may not be 0-mean during the learning process. We account for this by estimating mean of $\nu_t$ as well. We know

Figure 6: A comparison of speed of learning in Sparse Mountain Car, Puddle World and River Swim. In plots (a) and (b) lower on y-axis are better, whereas in (c) curves higher along y-axis are better. Sparse Mountain Car and Puddle World are episodic problems with a fixed experience budget. Thus the length of the lines in plots (a) and (b) indicate how many episodes each algorithm completed over 50,000 steps, and the height on the y-axis indicates the quality of the learned policy—lower indicates better performance. Note RLSVI did not show significant learning after 50,000 steps. The RLSVI result in Puddle World uses a budget of 1 million.

$\nu_t \sim \mathcal{N}(\bar{\nu}_t, \sigma_t^2)$. Therefore:

$$
\begin{aligned}
\bar{\nu}_{t+1} &= E[r_{t+1}] && - E[\mathbf{x}_t - \gamma \mathbf{x}_{t+1}]^\top \mathbf{w}_t \\
\bar{\nu}^2_{t+1} &= E[r_{t+1}^2] && - 2E[r_{t+1}(\mathbf{x}_t - \gamma \mathbf{x}_{t+1})]^\top \mathbf{w}_t \\
&&& + \mathbf{w}_t^\top E[(\mathbf{x}_t - \gamma \mathbf{x}_{t+1})(\mathbf{x}_t - \gamma \mathbf{x}_{t+1})^\top] \mathbf{w}_t
\end{aligned}
$$

These expected values are maintained incrementally. Utilizing this, $\sigma_{t+1}^2 = \bar{\nu}^2_{t+1} - \bar{\nu}_{t+1}^2$. We refer to Global variance UCB as GV-UCB. The algorithm is given in Algorithm 6.

### E.2 Bootstrapped upper confidence bounds

The strategy for action selection which utilizes bootstrapped confidence intervals, as proposed by White and White [46], is given in Algorithm 7. This action selection strategy can be used in conjunction with any learning algorithm to guide on-policy control. The algorithm requires a window of recent $\mathbf{w}$'s. The window can be maintained with a circular queue. The window is updated after each learning step of the main algorithm, resulting in a new $\mathbf{w}_t$ in the queue. The original UCBootstrap paper proposed both a global and a sparse updating mechanism, where only the global approach was theoretically justified. The sparse mechanism was used to reduce the number of parameters stored, particularly by taking advantage of tile-coding representations. We found in our experiments that the global approach worked just as well as the sparse approach, and so we include only the simpler, theoretically justified algorithm.

### E.3 DGPQ

Another approach to exploration is found in a model-free algorithm using gaussian processes named Delayed-GPQ (DGPQ) [8]. The pseudocode for DGPQ is in Algorithm 8. Any algorithm can be used to train the Gaussian processes, and for this paper we use the same algorithm as in [8]. The initialization of this algorithm requires the maximum reward and value, but for ease of use we transform the reward signal to $r_{new} = r - R_{max}$ so the means of the gaussian processes can be initialized to zero and $V_{max} = 0$.

A major problem with DGPQ is the large number of parameters needed to be set properly. Some intuition on setting these parameters can be found in [8] as well as in algorithm 8. As some guidance the width of the kernel determines how much a sample can generalize to other states, the thresholds $(\sigma_{tol}^2, \epsilon)$ determine how often we swap for new experience in the set basis vectors, and the Lipschitz constant $L_Q$ tunes the tradeoff between exploration and exploitation.

### E.4 LSPI-Rmax

LSPI-Rmax [19] combines LSPI [17] with Rmax [6] for online control in continuous state-spaces. Exploration is encouraged by determining the *knowness* of a transition, utilizing kernels. LSPI

**Algorithm 6** GV-UCB($\lambda$)

---

$\mathbf{A} \leftarrow \mathbf{0}, \mathbf{b} \leftarrow \mathbf{0}, \mathbf{z} \leftarrow \mathbf{0}, \mathbf{w} \leftarrow \mathbf{0},$
$\mathbf{B} \leftarrow \mathbf{I}, \mathbf{C} \leftarrow \mathbf{I}, \bar{\mathbf{z}} \leftarrow \mathbf{0}$
$p = 0.01, \eta = 10^{-4}, \beta = 0.001$
$\sigma = 1.0, \bar{r} = 0.0, \bar{r^2} = 100.0, \bar{\mathbf{d}} \leftarrow \mathbf{0}, \bar{\mathbf{d_r}} \leftarrow \mathbf{0}, \bar{\mathbf{D}} \leftarrow \mathbf{0}$
$\mathbf{x}_{s,\cdot} \leftarrow$ initial state-action features, for any action
$a \leftarrow$ GetOptimisticActionGlobal$(\mathbf{x}_{s,\cdot})$
**repeat**
    Take action $a$ and observe $\mathbf{x}_{s',\cdot}$ and $r$, and $\gamma$
    $a' \leftarrow$ GetOptimisticActionGlobal$(\mathbf{x}_{s',\cdot})$
    $\delta \leftarrow r + (\gamma \mathbf{x}_{s',a'} - \mathbf{x}_{s,a})^\top \mathbf{w}$
    $\mathbf{z} \leftarrow \gamma\lambda\mathbf{z} + \mathbf{x}_{s,a}$
    $\mathbf{b} \leftarrow (1 - \beta)\mathbf{b} + \beta r\mathbf{z}$
    $\mathbf{A} \leftarrow (1 - \beta)\mathbf{A} + \beta\mathbf{z}(\mathbf{x}_{s,a} - \gamma\mathbf{x}_{s',a'})^\top$
    $\triangleright$ Update $\mathbf{C}$
    $\bar{\mathbf{z}} \leftarrow (1 - \beta)\bar{\mathbf{z}} + \beta\mathbf{z}$
    $\mathbf{a} \leftarrow \mathbf{B}^\top\bar{\mathbf{z}}$
    **for** $i$ such that $\mathbf{z}_i \neq 0$ **do**
        **for** $j$ such that $\mathbf{z}_j \neq 0$ **do**
            $\mathbf{C}_{ij} \leftarrow (1 - \beta)\mathbf{C}_{ij} + \beta\mathbf{a}_i\mathbf{a}_j$
    $\triangleright$ Update $\sigma$
    $\bar{r} \leftarrow (1 - \beta)\bar{r} + \beta r$
    $\bar{r^2} \leftarrow (1 - \beta)\bar{r^2} + \beta r^2$
    $\bar{\mathbf{d}} \leftarrow (1 - \beta)\bar{\mathbf{d}} + \beta(\mathbf{x}_{s,a} - \gamma\mathbf{x}_{s',a'})$
    $\bar{\mathbf{d_r}} \leftarrow (1 - \beta)\bar{\mathbf{d_r}} + \beta r(\mathbf{x}_{s,a} - \gamma\mathbf{x}_{s',a'})$
    $\bar{\mathbf{D}} \leftarrow (1 - \beta)\bar{\mathbf{D}} + \beta(\mathbf{x}_{s,a} - \gamma\mathbf{x}_{s',a'})(\mathbf{x}_{s,a} - \gamma\mathbf{x}_{s',a'})^\top$
    $\bar{\nu} = \bar{r} - \bar{\mathbf{d}}^T\mathbf{w}$
    $\bar{\nu^2} = \bar{r^2} - 2\bar{\mathbf{d_r}}^T\mathbf{w} + \mathbf{w}^\top\bar{\mathbf{D}}\mathbf{w}$
    $\sigma = \sqrt{\bar{\nu^2} - \bar{\nu}^2}$
    $\triangleright$ Update $\mathbf{w}$ and $\mathbf{B} \approx \mathbf{A}^{-\top}$
    $\alpha = \min\left\{1.0, \frac{0.01}{||\mathbf{A}||_F^2||\mathbf{x}_{s,a}||_2^2 + 1.0}\right\}$
    $\mathbf{B} \leftarrow \mathbf{B} - \alpha\mathbf{A}(\mathbf{A}^\top\mathbf{B}\mathbf{x}_{s,a} - \mathbf{x}_{s,a})\mathbf{x}_{s,a}^\top$
    $\mathbf{w} \leftarrow \mathbf{w} + (\mathbf{B} + \eta\mathbf{I})(\mathbf{b} - \mathbf{A}\mathbf{w})$
    $\mathbf{x}_{s,a} \leftarrow \mathbf{x}_{s',a'}$    and    $a \leftarrow a'$
**until** agent done interaction with environment

---

algorithm is designed for a batch setting, where the LSTD solution is computed in closed form for staged batches of data. However, because it accumulates optimistic values, it can be simply converted into an online algorithm using incremental updates to the matrix $\mathbf{A}$ and $\mathbf{b}$, as done in Li et al. [19].

We summarize this extension in pseudocode as Algorithm 10. Until states become known, the algorithm estimates action-values that predict the maximum possible return; once a state becomes known, it starts to use actual rewards sampled from the environment. To estimate the *knowness* of a state under function approximation, we use feature counts. Each state has a set of active features; the active feature with the minimum count reflects an upper bound on the number of times that this state has been seen. Once a states active features have been seen frequently enough, it becomes known.

## E.5 RLSVI

RLSVI [31] is an algorithm that maintains a distribution over the possible value functions. The value functions are assumed to be linearly parametrized. While the main algorithm proposed uses a finite-horizon assumption, a modified version proposed in the Appendix of the paper does not, and this is the version used in the experiments here.

**Algorithm 7** UCBootstrap($\mathbf{x}_{s,.}$) select action from state features $\mathbf{x}_{s,.}$ at time $t$

---

$l$ = block length, $B$ = number of bootstrap resamples, $w$ = number (window) of value functions weights to store and confidence level $\alpha$

examples: $l = 10$, $B = 50$, $w = 100$, $\alpha = 0.05$

$\quad M \leftarrow \lfloor w/l \rfloor$ $\qquad\qquad$ ▷ num of length $l$ blocks to sample with replacement and concatenate

$\quad$ **for** each action $a$ **do**

$\qquad Q_N \leftarrow \{\mathbf{w}_{t-w}^\top \mathbf{x}_{s,a}, \ldots, \mathbf{w}_{t-1}^\top \mathbf{x}_{s,a}\}$

$\qquad \bar{Q}_N \leftarrow \text{mean}(Q_N)$ $\qquad$ ▷ The mean value for this $(s,a)$, given the window of recent weights

$\qquad$ Blocks = $\Big\{ \{[Q_N[0], \ldots, Q_N[l\text{-}1]\}, \{[Q_N[1], \ldots, Q_N[l]\},$

$\qquad\qquad\qquad\qquad \ldots, [Q_N[w\text{-}l], \ldots, Q_N[w\text{-}1]] \Big\}$

$\qquad$ **for all** $i = 1$ to $B$ **do**

$\qquad\qquad$ **for all** $j = 1$ to $M$ **do**

$\qquad\qquad\qquad A_j^* \leftarrow$ random block from Blocks (chosen with replacement)

$\qquad\qquad A \leftarrow (A_1^*, A_2^*, \ldots, A_M^*)$ $\qquad\qquad\qquad\qquad\qquad$ ▷ Concatenate blocks

$\qquad\qquad T_i^* = \frac{1}{lM} \sum_{k=1}^{lM} A[k]$ $\quad$ ▷ $i$th bootstrap estimate is the mean of the $M$ concatenated blocks

$\qquad T \leftarrow \text{sort}(\{T_1^*, \ldots, T_B^*\})$ $\qquad\qquad\qquad\qquad\qquad$ ▷ ascending order

$\qquad j \leftarrow \lfloor \frac{B\alpha}{2} + \frac{\alpha+2}{6} \rfloor$ $\quad$ ▷ $j$ is the position of the critical samples to help estimate the continuous sample quantile

$\qquad r \leftarrow \frac{B\alpha}{2} + \frac{\alpha+2}{6} - j$ $\qquad\qquad\qquad\qquad\qquad\qquad$ ▷ $r$ is the remainder

$\qquad T_{\alpha/2}^* \leftarrow (1-r)T_j^* + rT_{j+1}^*$ $\qquad\qquad\qquad\qquad\qquad$ ▷ the $\alpha/2$ sample quantile

$\qquad u_a \leftarrow 2\bar{Q}_N - T_{\alpha/2}^*$

$\quad a = \text{argmax}_{a\in\mathcal{A}} u_a$

$\quad$ **return** $a$

---

## F  Alternative updates for LSTD

The update for $\mathbf{w}$ using $\mathbf{A}$ and $\mathbf{b}$ in UCLS is the result of an empirical investigation into alternative linear system solvers. We investigated using a Sherman-Morrison update, with exponential averaging (in Algorithm 11) as well as improved incremental inverse updates, including one for pseudo-inverses [23]. This update has a confounding role for $\eta$, and for small $\eta$ we found it less stable than our proposed update. We investigated iterative updates with a fixed stepsize, $\mathbf{w}_{t+1} = \mathbf{w}_t + \alpha(\mathbf{b}_t - \mathbf{A}_t\mathbf{w})$; the addition of the step-size, however, removes some of the parameter-free benefits of LSTD. We investigated conjugate gradient updates, as in Algorithm 12. We finally derived the iterative update proposed, for $\mathbf{B} \approx \mathbf{A}^{-\top}$, to obtain a preconditioner for the iterative update.

For completeness, we include the derivation for the Sherman-Morrison update. The derivation for $\mathbf{A}_{t+1}^{-1}$ using $\mathbf{A}_{t+1} = (1-\beta)\mathbf{A}_t + \beta uv^T$ is as follows:

$$\mathbf{A}_t^{-1}\mathbf{A}_{t+1} = (1-\beta)\mathbf{I} + \beta\mathbf{A}_t^{-1}uv^T$$

Converting it to terms of $\mathbf{A}_{t+1}^{-1}$:

$$\mathbf{A}_{t+1}^{-1} = ((1-\beta)\mathbf{I} + \beta\mathbf{A}_t^{-1}uv^T)^{-1}\mathbf{A}_t^{-1}$$

$$= \left( \frac{1}{(1-\beta)}\mathbf{I} + \frac{1}{1 + \frac{\beta}{1-\beta}v^T\mathbf{A}_t^{-1}u} \frac{\beta}{(1-\beta)^2}\mathbf{A}_t^{-1}uv^T \right)\mathbf{A}_T^{-1}$$

$$= \frac{1}{(1-\beta)}\mathbf{A}_t^{-1} + \frac{\frac{\beta}{(1-\beta)}\mathbf{A}_t^{-1}uv^T\mathbf{A}_t^{-1}}{(1-\beta) + \beta v^T\mathbf{A}_t^{-1}u}$$

The first step utilizes a Lemma in [24].

## G  Extended results

We show additional results here comparing UCLS to Sarsa with optimistic initialization and GV-UCB (p=0.5), Figure 7; along with DGPQ in MCSparse. Also included are plots showing best and worst

**Algorithm 8** DGPQ($k(\cdot, \cdot), d(\cdot, \cdot), L_Q, Env, \mathcal{A}, R_{max}, s_0, \gamma, \sigma^2, \sigma^2_{tol}, \epsilon$)

$k(\cdot, \cdot), d(\cdot, \cdot)$ are typically the RBF w/ bandwidth = $\sigma^2$ and euclidean distance respectively.
$L_Q$ correlates with exploration.
$\mathcal{A}$ is the set of possible actions.
$\gamma$ is the discount factor.
$\sigma^2_{tol}$ is the tolerance of induced variance of using a new point to update a GP
Found useful ranges for parameters during sweeps:
$\sigma^2 \in [0.001, 0.5], \sigma^2_{tol} \in [0.01, 0.1], \epsilon \in [0.01, 0.1], L_Q \in [1, 20]$

1: $\hat{Q}(s,a) \overset{\text{def}}{=} \min($
$\qquad V_{max},$
$\qquad \min_{(s_i,a) \in \hat{Q}_a.BV} \{[\hat{\mu}_i + L_Q d((s,a), (s_i, a))]\}$
$\qquad )$
2: **for** $a \in A$ **do**
3: $\quad \hat{Q}_a.BV = \emptyset$
4: $\quad GP_a = GP.init(\mu = \frac{R_{max}}{1-\gamma}, k(\cdot, \cdot))$
5: **for** $t \in [0, T]$ **do**
6: $\quad a_t = \operatorname{argmax}_a \hat{Q}(s,a)$
7: $\quad$ //take action $a_t$ in state $s_t$, observe $(s_{t+1}, r_t)$
8: $\quad (s_{t+1}, r_t) = Env(s_t, a_t)$
9: $\quad q_t = r_t + \gamma \max_a \hat{Q}(s_{t+1})$
10: $\quad \sigma^2_1 = GP_{a_t}.variance(s_t)$
11: $\quad$ //If the new sample is not well covered by $GP_{a_t}$
12: $\quad$ **if** $\sigma^2_1 > \sigma^2_{tol}$ **then**
13: $\qquad GP_{a_t}.update(s_t, q_t)$
14: $\quad \sigma^2_2 = GP_{a_t}.variance(s_t)$
15: $\quad$ //If the $GP_{a_t}$ now well covers a previously unknown state and the new approximation is $2\epsilon$ less than what is found in $\hat{Q}$ (i.e. is a less optimistic estimate).
16: $\quad$ **if** $\{\sigma^2_1 > \sigma^2_{tol} \geq \sigma^2_2\}$ **and**
$\qquad \{\hat{Q}_{a_t}(s_t) - GP_{a_t}.mean(s_t) > 2\epsilon\}$ **then**
17: $\qquad \mu = GP_{a_t}.mean(s_t) + \epsilon$
18: $\qquad \hat{Q}_{a_t}.BV.add((s_t, a_t), \mu)$
19: $\qquad$ **for** $((s_j, a_t), \mu_j) \in \hat{Q}_{a_t}.BV$ **do**
20: $\qquad\quad$ **if** $\mu_j \leq \mu + L_Q d((s_t, a_t), (s_j, a_t))$ **then**
21: $\qquad\qquad \hat{Q}_{a_t}.BV.delete(((s_j, a_t), \mu_j))$
22: $\qquad$ //To prevent slow learning or halted learning reset the current GPs and initialize to the current estimates.
23: $\qquad \forall a \in A, GP_a = GP.init(\hat{\mu} = \hat{Q}_a, k(\cdot, \cdot))$

---

**Algorithm 9** IsKnown($s, a$)

1: // Uses the minimum count of the features for a state, to decide if $s, a$ is known
2: // If $a$ not given, sums over all $a$
3: $m = 5$
4: **if** $a$ not given **then**
5: $\quad \mathbf{f} \leftarrow \sum_a \mathbf{c}(\mathbf{x}_{s,a}) \in \mathbb{R}^d$
6: **else**
7: $\quad \mathbf{f} \leftarrow \mathbf{c}(\mathbf{x}_{s,a}) \in \mathbb{R}^d$
8: **if** $\min(\mathbf{f}) > m$ **then**
9: $\quad$ **return** "Known"
10: **else**
11: $\quad$ **return** "Not Known"

**Algorithm 10** Incremental LSPI-Rmax($m$)

1: $\mathbf{A} \leftarrow \mathbf{0}, \mathbf{b} \leftarrow \mathbf{0}, \mathbf{z} \leftarrow \mathbf{0}, \mathbf{w} \leftarrow \mathbf{0}$,
2: $\mathbf{B} \leftarrow \mathbf{I}, \mathbf{c} \leftarrow \mathbf{0}$
3: $\eta = 10^{-4}, \beta = 0.001, \lambda = 0, G_{\max} = r_{\max}/(1-\gamma)$ if continuing or $\gamma \neq 1$, else $G_{\max} = r_{\max}h$ for a predicted maximum episode length (e.g., $h = 10000$).
4: $\mathbf{x}_{s,\cdot} \leftarrow$ initial state-action features, for any action
5: $a \leftarrow$ greedy action according to value estimates given by $\mathbf{x}_{s,a}^\top \mathbf{w}$
6: **repeat**
7:     Take action $a$ and observe $\mathbf{x}_{s',\cdot}$ and $r$, and $\gamma$
8:     $a' \leftarrow$ greedy action according to value estimates given by $\mathbf{x}_{s',a'}^\top \mathbf{w}$
9:     $\mathbf{z} \leftarrow \gamma\lambda\mathbf{z} + \mathbf{x}_{s,a}$
10:     **if** IsKnown$(s, a)$ **then**
11:         **if** IsKnown$(s')$ **then**
12:             $\mathbf{A} \leftarrow (1-\beta)\mathbf{A} + \beta\mathbf{z}(\mathbf{x}_{s,a} - \gamma\mathbf{x}_{s',a'})^\top$
13:             $\mathbf{b} \leftarrow (1-\beta)\mathbf{b} + \beta r\mathbf{z}$
14:         **else**
15:             $\mathbf{A} \leftarrow (1-\beta)\mathbf{A} + \beta\mathbf{x}_{s,a}\mathbf{x}_{s,a}^\top$
16:             $\mathbf{b} \leftarrow (1-\beta)\mathbf{b} + \beta(r + \gamma G_{\max})\mathbf{x}_{s,a}$
17:     **else**
18:         $\mathbf{A} \leftarrow (1-\beta)\mathbf{A} + \beta\mathbf{x}_{s,a}\mathbf{x}_{s,a}^\top$
19:         $\mathbf{b} \leftarrow (1-\beta)\mathbf{b} + \beta G_{\max}\mathbf{x}_{s,a}$
20:     **for** $\forall \tilde{a} \in A\backslash a$ **do**
21:         **if** !IsKnown$(s, \tilde{a})$ **then**
22:             $\mathbf{A} \leftarrow (1-\beta)\mathbf{A} + \beta\mathbf{x}_{s,\tilde{a}}\mathbf{x}_{s,\tilde{a}}^\top$
23:             $\mathbf{b} \leftarrow (1-\beta)\mathbf{b} + \beta G_{\max}\mathbf{x}_{s,\tilde{a}}$
24:     $\mathbf{c} \leftarrow \mathbf{c} + \mathbf{x}_{s,a}$
25:     $\alpha = \min\left\{1.0, \frac{0.01}{||\mathbf{A}||_F^2||\mathbf{x}_{s,a}||_2^2+1.0}\right\}$
26:     $\mathbf{B} \leftarrow \mathbf{B} - \alpha\mathbf{A}(\mathbf{A}^\top\mathbf{B}\mathbf{x}_{s,a} - \mathbf{x}_{s,a})\mathbf{x}_{s,a}^\top$
27:     $\mathbf{w} \leftarrow \mathbf{w} + (\mathbf{B} + \eta\mathbf{I})(\mathbf{b} - \mathbf{A}\mathbf{w})$
28:     $\mathbf{x}_{s,a} \leftarrow \mathbf{x}_{s',a'}$   and  $a \leftarrow a'$
29: **until** agent done interaction with environment

---

**Algorithm 11** LSTD($\lambda$) with Sherman-Morrison update

1: $\mathbf{A}^{-1} \leftarrow \frac{1}{\eta}\mathbf{I}, \mathbf{b} \leftarrow \mathbf{0}, \mathbf{z} \leftarrow \mathbf{0}, \mathbf{w} \leftarrow \mathbf{0}$,
2: $\mathbf{x}_{s,\cdot} \leftarrow$ initial state-action features, for any action
3: $a \leftarrow \epsilon$-greedy action according to value estimates given by $\mathbf{x}_{s,a}^\top\mathbf{w}$
4: **repeat**
5:     Take action $a$ and observe $\mathbf{x}_{s',\cdot}$ and $r$, and $\gamma$
6:     $a' \leftarrow \epsilon$-greedy action according to value estimates given by $\mathbf{x}_{s',a'}^\top\mathbf{w}$
7:     $\mathbf{z} \leftarrow \gamma\lambda\mathbf{z} + \mathbf{x}_{s,a}$
8:     $\beta = \frac{1}{t}$
9:     $\mathbf{b} \leftarrow \mathbf{b} + \beta(r\mathbf{z} - \mathbf{b})$
10:     $\mathbf{v} \leftarrow \left((\mathbf{x}_{s,a} - \gamma\mathbf{x}_{s',a'})^\top\mathbf{A}^{-1}\right)^\top$
11:     $\mathbf{A}^{-1} \leftarrow \frac{1}{(1-\beta)}\mathbf{A}^{-1} + \frac{\frac{\beta}{(1-\beta)}\mathbf{A}^{-1}\mathbf{z}\mathbf{v}^\top}{(1-\beta)+\beta\mathbf{v}^\top\mathbf{z}}$
12:     $\mathbf{w} \leftarrow \mathbf{A}^{-1}\mathbf{b}$
13:     $\mathbf{x}_{s,a} \leftarrow \mathbf{x}_{s',a'}$   and  $a \leftarrow a'$
14: **until** agent done interaction with environment

---

**Algorithm 12** LSTD($\lambda$) with Conjugate Gradient

---

$\mathbf{A} \leftarrow \mathbf{0}, \mathbf{b} \leftarrow \mathbf{0}, \mathbf{z} \leftarrow \mathbf{0}, \mathbf{w} \leftarrow \mathbf{0},$
$\mathbf{x}_{s,\cdot} \leftarrow$ initial state-action features, for any action
$a \leftarrow \epsilon$-greedy action according to value estimates given by $\mathbf{x}_{s,a}^\top \mathbf{w}$
**repeat**
    Take action $a$ and observe $\mathbf{x}_{s',\cdot}$ and $r$, and $\gamma$
    $a' \leftarrow \epsilon$-greedy action according to value estimates given by $\mathbf{x}_{s',a'}^\top \mathbf{w}_{a'}$
    $\mathbf{z} \leftarrow \gamma\lambda\mathbf{z} + \mathbf{x}_{s,a}$
    $\beta_t = \frac{1}{t}$                                        $\triangleright$ or constant such as $\beta_t = 0.01$
    $\mathbf{b} \leftarrow \mathbf{b} + \beta_t(r\mathbf{z} - \mathbf{b})$
    $\mathbf{A} \leftarrow \mathbf{A} + \beta_t(\mathbf{z}(\mathbf{x}_{s,a} - \gamma\mathbf{x}_{s',a'})^\top - \mathbf{A})$
    $\mathbf{w} \leftarrow$ ConjugateGradient($\mathbf{A} + \eta\mathbf{I}, \mathbf{b}, \mathbf{w}, \eta_r$)
    $\mathbf{x}_{s,a} \leftarrow \mathbf{x}_{s',a'}$    and    $a \leftarrow a'$
**until** agent done interaction with environment

---

---

**Algorithm 13** Conjugate Gradient($\mathbf{A}, \mathbf{b}, \mathbf{w}, \eta_r$)

---

1: tol = 0.001
2: $\tilde{\mathbf{A}} = \mathbf{A}^T\mathbf{A} + \eta_r\mathbf{I}$
3: $\mathbf{r} \leftarrow \mathbf{b} - \tilde{\mathbf{A}}\mathbf{w}$
4: $\mathbf{d} \leftarrow \mathbf{r}$
5: **repeat**
6:     $\alpha \leftarrow \frac{\mathbf{r}^\top\mathbf{r}}{\mathbf{d}^\top\mathbf{A}\mathbf{d}}$
7:     $\mathbf{w} \leftarrow \mathbf{w} + \alpha\mathbf{d}$
8:     $\mathbf{r}' \leftarrow \mathbf{r} - \alpha\tilde{\mathbf{A}}\mathbf{d}$
9:     $\beta \leftarrow \frac{\mathbf{r}'^\top\mathbf{r}'}{\mathbf{r}^\top\mathbf{r}}$
10:     $\mathbf{d} \leftarrow \mathbf{r}' + \beta\mathbf{d}$
11: **until** CG converged ($||\mathbf{r}'||_2^2 \leq$ tol) or a fixed number of steps reached
12: **return w**

---

Figure 7: Learning curves in the three domains comparing UCLS to additional methods. In the first two plots lower on y-axis indicates better performance, whereas in the right-most plot higher along y-axis is better.

runs for UCLS and DGPQ — the two closest competitors — to show the variance of each algorithm Figure 8. Additionally, to empirically reinforce the utility of contextual confidence interval radius (CIR) over global CIR, we evaluate the policies obtained by UCLS and GV-UCB after 50,000 learning samples in River Swim and present the results in Figure 9.

As mentioned in the main results, Sarsa with optimistic initialization performs remarkably well in these domains. In Sparse Mountain Car, as DGPQ converts the sparse reward dynamics to a dense one, it outperforms Sarsa with optimistic initialization as well. In Puddle World, UCLS matches up to Sarsa's policy. In River Swim UCLS experiences minimal regret when compared to Sarsa's control policy. With the loss of contextual variance estimates GV-UCB explores the complete state space more thoroughly, and therefore performs poorly. The explicit upper confidence bound given by

UCLS does not suffer from this, and sufficiently explores the domain to converge to an optimal policy without excessively exploring. For regions where there is low variance, the upper-confidence-bound converges more quickly to zero, whereas it remains higher in regions of uncertainty. Therefore, contextual variance estimates provide the flexibility of variable convergence based on the variance of the region, and global variance estimates decay too slowly. When the policies obtained by GV-UCB and UCLS are evaluated, it is clear that the policy obtained by UCLS is much closer to the optimal policy than the policy obtained by GV-UCB, showing that the exploration strategy used by UCLS is more *data efficient*.

Figure 8: Best and worst run curves for DGPQ(top) and UCLS(bottom). From left to right: Sparse Mountain Car, Puddle World, River Swim.

Figure 9: Policy evaluation plots comparing variations of final policy obtained by UCLS ($p = 0.1$) and GV-UCB ($p = 10e{-}5$) after 50,000 learning steps in River Swim. Policies with (M) indicate greedy policy w.r.t. mean estimates, whereas policies with (M+CIR) indicate greedy policies w.r.t. (mean + CIR) estimates. In the left plot it can be seen that UCLS(M) and UCLS(M+CIR) perform almost as well as the optimal policy, whereas both versions of GV-UCB are still sub-optimal in many parts of the state space. Additionally, the overlap of UCLS(M) and UCLS(M+CIR) indicates that contextual CIR fades faster than global CIR, and is a more data-efficient exploration strategy. The right plot helps contrast the final policies obtained to the actual control policy used during learning (indicated by just UCLS and GV-UCB).