[Reviews · NeurIPS 2018]

Reviewer 1



PAPER SUMMARY The authors propose a value-based reinforcement learnig (RL) method based on least-squares temporal difference learning (LSTD). Concretely, they find upper bounds on the action-values learned by LSTD under a fixed policy. These bounds can then be used by an optimistic approach (upper confidence bound (UCB)) to guide exploration in a data-efficient manner. The authors illustrate the practical benefits of the proposed method by comparing it to a number of other methods on a range of simple simulated RL problems. REVIEW The paper is written in a clear and rigorous fashion. The derived bound is theoretically interesting and its practical benefits have also been illustrated. The experimental tasks are relatively simple, but I think this is acceptable given the theoretical nature of this paper. Since I am not an expert in value-based RL, it is not completely clear to me how the proposed method compares to related work, but as far as I can judge this article would constitute a valuable publication.

Reviewer 2



The paper establishes an upper-confidence bound for the policy evaluation task using a least-squares TD approach. The bound is then used for optimistic exploration to solve the control problem. The proposed approach is definitely interesting and novel to my knowledge. The derived upper-confidence bounds are only valid for policy evaluation, but they are used to solve the control problem. Therefore there's a gap between theory and practice, which is acknowledged by the authors, but the whole discussion about how to address with heuristics the gap between the validity of the derived upper-confidence bounds (for a fixed policy) and its use in a policy iteration scheme appears in appendix C. I think the paper would be clearer if it were reorganized. The discussion in appendix C is important in my opinion, and should therefore appear in the main paper. Besides, the description of UCLS should also be in the main part, as this is the algorithm evaluated in the experimental section. The proofs of Th.1 and Cor.1 could be in the appendix. Minor remarks and typos: l.94: directlty l.129: When make (2): should r_{t+1} be R_{t+1}? (4): it would be clearer to specify that it is whp l.178: upper -> lower? l.203: are constitute p.6: it's -> its l.230, 235: the names of the authors of [44] shouldn't appear l.248: why are the averages computed on different number of runs?

Reviewer 3



Summary: In this paper, the authors have described how context dependent, i.e. state and action pair dependent, Upper Confidence Bounds can be calculated for the Least Squares Temporal Difference (LSTD) Learning algorithm. They argue that this is a principled way of exploring the interactions with the environment because is specially important for least square methods because other strategies are either inefficient (e-greedy) or cannot be used in a meaningful manner (optimistic initialization). The computation of the upper confidence part is interesting and deviates from the standard proofs for Multi-arm bandit cases because the samples in case of LSTD are not independent of each other. The authors address this issue by imposing a bound using the (weaker) Chebychev's inequality instead of the Bernstein's inequality. In their evaluation, they use a bound on the total compute rather than on the number of episodes. Though non-standard, the evaluation procedure did make sense for the claims they make that their algorithm is able to attain better performance with fewer steps wasted exploring. The authors have provided pseudo-code for all their baselines and, hopefully, would make the source code for their experiments available, making the paper reproducible. Key Contributions: - First principled method for using optimistic Upper Confidence Bound algorithms with LSTD. - Interesting use of Chebychev's inequality to handle the non-IID setting and allowing updates in an online fashion for RL. The paper is clearly written and is easy to follow. The problem is interesting and their algorithm fills in a gap in the space of algorithms we have in RL. The derivation is novel through relatively straight forward, though some clarifications which have been relegated to the appendix ought to appear in the main body (see below). The baselines that the authors have chosen to compare with are competitive and their description of the experimental setup is exhaustive. The future directions of work also looks promising, and may lead to significant advances in the future. That said, there are a few minor issues with the work. The regret analysis of their approach is preliminary at best, though, as stated, the future directions look promising. Some heuristics in their algorithms (e.g. handling third issue discussed in appendix C) explicitly depend on the tile coding of the states/actions. It is unclear how easy it will be to design similar heuristics in case a non-linear embedding of the input state is used (e.g. using NNs). Authors mention this as a potential direction for future work, but a comment about the reliance of the implementation on this particular encoding of the state in the main-text is warranted. However, the writing/presentation of the proof should be improved. For example, in line 165, the term "This estimation" is ambiguous. The equation after line 167 does not seem to follow from the equation presented above, but rather directly from the equation above line 165. Also, the last inequality above line 175 is true only with high probability, which is not clearly mentioned. The relationship of c√(a^2 + b^2) to the previous equation can also use more explanation. It is also important to state in the last paragraph of the section (instead of in the appendix) that since the estimates for noises are *not* available, the TD error will be used as the proxy. As this detail is somewhat of a departure from the multi-arm bandit setting, it would be useful for readers familiar with that domain. Overall, I believe that the paper makes an interesting contribution and should be accepted for publication. Minor issues: - line 94: "direclty" - line 129: "When" -> "we" - line 203: "are constitute" - line 205: "high-lighted" -> "highlighted" - line 229: "it's" -> "its" - "River Swim" task does not describe the termination condition/goal state in the description. - line 280: "the the" - lines 285 -- 287: the paragraph is out of place. - line 289: "a sound upper bounds": unconventional use of the term 'sound'. - Error bars are missing from several figures (notably, 3(c) and 5). - Figure 3 caption: LSTD-CG-... are not defined. Also in Figure 5 caption, CG seems to be used without prior definition. - Is it possible to also show somehow how quickly the confidence bounds for various states decrease and how it is related to the value of the state? It would be interesting to see actually how much exploration effort the algorithms wastes on low valued states. ---- Update: Having read the authors' response, I would recommend that the explanations/changes they have proposed for (1),(2) in the minor changes section ought to be included in the main paper. Overall, my opinion about the paper remains unchanged.